# Determinants of genome-wide distribution and evolution of uORFs in eukaryotes

Hong Zhang [1], Yirong Wang[1,2], Xinkai Wu [1], Xiaolu Tang [1], Changcheng Wu [1] & Jian Lu [1✉]

Upstream open reading frames (uORFs) play widespread regulatory functions in modulating mRNA translation in eukaryotes, but the principles underlying the genomic distribution and evolution of uORFs remain poorly understood. Here, we analyze ~17 million putative canonical uORFs in 478 eukaryotic species that span most of the extant taxa of eukaryotes. We demonstrate how positive and purifying selection, coupled with differences in effective population size ($N_e$), has shaped the contents of uORFs in eukaryotes. Besides, gene expression level is important in influencing uORF occurrences across genes in a species. Our analyses suggest that most uORFs might play regulatory roles rather than encode functional peptides. We also show that the Kozak sequence context of uORFs has evolved across eukaryotic clades, and that noncanonical uORFs tend to have weaker suppressive effects than canonical uORFs in translation regulation. This study provides insights into the driving forces underlying uORF evolution in eukaryotes.

[1] State Key Laboratory of Protein and Plant Gene Research, Center for Bioinformatics, School of Life Sciences, Peking University, Beijing, China. [2] College of Biology, Hunan University, Changsha, China. ✉email: LUJ@pku.edu.cn

Upstream open reading frames (uORFs) are short open reading frames (ORFs) that have start codons located in the 5′ untranslated regions (UTRs) of eukaryotic mRNAs. uORFs can attenuate the translational initiation of downstream coding sequences (CDSs) by sequestering or competing for ribosomes[1–10]. For an AUG triplet in the 5′ UTR (defined as "uAUG" hereafter), it can function as the start codon of a uORF that has a stop codon either preceding the start codon of the downstream CDS (nonoverlapping uORF, nORF) or residing in the body of the downstream CDS (out-of-frame overlapping uORF, oORF)[4,11–18]. Less frequently, an uAUG can function as the start codon of an ORF whose stop codon overlaps with the stop codon of the downstream CDS (N-terminal extension, NTE)[4,19–21]. The advent of ribosome profiling[22–26], a method that determines the ribosome occupancy on mRNAs at the codon level, has enabled the genome-wide characterization of uORFs and NTEs that showed evidence of translation in various species with high sensitivity and accuracy[12,16,17,27–36]. Besides the canonical uORFs (beginning with an AUG start codon and ending with a UAA/UAG/UGA stop codon), the modified ribosome profiling methods[4,37], which detect initiating ribosomes in cells treated with harringtonine[32,34] or lactimidomycin[38–40], have provided further evidence showing that many noncanonical uORFs (beginning with a non-AUG codon and ending with a UAA/UAG/UGA stop codon) might be prevalent and functionally important. Collectively, recent studies have demonstrated that uORFs are prevalently translated in eukaryotic cells and that uORF-mediated regulation plays important roles in tuning the translational program during development[32,41–45] or stress responses[10,27,46–55].

It is well accepted that canonical uORFs are generally deleterious and are depleted in the 5′ UTRs of eukaryotic genomes[56–60], and mutations that generate polymorphic uORFs are also usually deleterious and selected against in humans[19,61–66] and flies[32]. Nevertheless, our recent study indicated that many uAUGs recently fixed in *Drosophila melanogaster* were driven by positive Darwinian selection[32], which suggests that some uORFs and NTEs might be adaptive. Despite these exciting progress, the principles underlying the genomic distribution and evolution of uORFs and NTEs are poorly understood. For example, the following questions remained unanswered: (1) What is the role of natural selection in shaping the genome-wide contents of uORFs and NTEs in eukaryotes at the micro- and macroevolutionary scales? (2) Can we detect signatures of positive selection on uORFs and NTEs in clades other than *Drosophila*? (3) Are the sequence characteristics that influence the efficacy of uORF-mediated translational repression conserved between different eukaryotic species? Answers to these questions will not only help elucidate the role of translational regulation in adaptation, but also advance our understanding of the mechanisms underlying protein homeostasis in health and disease.

Here, we systematically characterize 16,907,129 uAUGs in 478 eukaryotic species and explore various factors and forces that determine the genome-wide distributions of uORFs and NTEs across genes and species. Our results suggest that differences in uORF occurrences across genes are mainly influenced by gene expression levels, while the interspecific variability of uORFs is shaped by the effective population size ($N_e$). We also compare the conservation patterns of start codons versus coding regions of the canonical uORFs in different clades, disentangled the relationship between the Kozak sequence context and the translational efficiency of uORFs, and explore the evolution of Kozak contextual characteristics across eukaryotes. Our analyses present a broad overview of the interspecies variability of uAUGs in eukaryotes and provide insights into the general principles underlying the distribution and sequence evolution of uORFs and NTEs in eukaryotes.

## Results

### Characterization of putative canonical uORFs and NTEs in 478 eukaryotes.

We developed a bioinformatic pipeline and characterized uAUGs in the genomes of 478 eukaryotic species, including 242 fungi, 20 protists, and 216 multicellular eukaryotes that comprise plants and animals. As most species surveyed in this study currently have no ribosome profiling data, and it is very challenging to predict the noncanonical uORFs in silico reliably, we only focused on the putative canonical uORFs that start with the AUG start codon. In what follows, the uORFs analyzed in this study are restricted to the putative canonical uORFs unless explicitly stated otherwise (all the annotated uORFs and NTEs are presented in figshare[67]).

The number of annotated protein-coding genes in the 242 fungi ranged from 3623 (*Pneumocystis murina*) to 32,847 (*Fibularhizoctonia sp.*). A total of 3,469,095 uAUGs were identified in these fungal genomes, with the number ranging from 1233 (*Malassezia sympodialis*) to 94,695 (*Verticillium longisporum*) (Supplementary Data 1). Since many protists use alternative nuclear genetic codes involving stop-codon reassignments[68–73] or obligatory frameshifting at internal stop codons[74], here we only focused on 20 protists that use the standard genetic code (Supplementary Data 1). Among the 20 protists, the number of annotated protein-coding genes ranged from 5389 (*Plasmodium vivax*) to 38,544 (*Emiliania huxleyi*), and the number of uAUGs ranging from 1903 (*Plasmodium falciparum*) to 99,859 (*Cystoisospora suis*), which resulted in a total of 391,565 uAUGs in these protist genomes (Supplementary Data 1).

The 216 multicellular plants and animals, whose last common ancestor was dated to 1.5 billion years ago[75], span the following taxa: (1) 41 plants, including mosses, eudicotyledons, and monocotyledons; (2) 38 invertebrates, including sponges, ctenophores, flatworms, cephalopods, mites, centipedes, crustaceans, springtails, insects, and tunicates; and (3) 137 vertebrates, including hagfishes, fishes, coelacanths, toads, lizards, turtles, crocodiles, birds, and mammals (Fig. 1). Among these species, mammals, including monotremes ($n = 1$), marsupials ($n = 3$), an armadillo ($n = 1$), laurasiatherians ($n = 17$), a rabbit ($n = 1$), rodents ($n = 21$), and primates ($n = 24$), constituted the largest clade ($n = 68$ species). Nematodes were excluded from the analyses because *trans*-splicing alters the 5′ UTR sequences of many mRNAs in their transcriptomes[45,76].

The number of annotated protein-coding genes in the 216 multicellular plants and animals ranged from 10,581 (*Bombus terrestris*) to 107,545 (*Triticum aestivum*), and 3388 (*Schistosoma mansoni*) to 68,741 (*T. aestivum*) of these genes exhibited annotated 5′ UTRs for at least one transcript (Fig. 1a and Supplementary Data 1). In these species, the annotated 5′ UTRs were usually shorter than 500 nt (the median length of the annotated 5′ UTRs ranged from 22 nt in *Arabidopsis lyrata* to 477 nt in *Physcomitrella patens*; Supplementary Fig. S1 and Supplementary Data 1). The number of uAUGs ranged from 3249 (*Drosophila willistoni*) to 798,433 (*Theobroma cacao*). Altogether, we identified a total of 13,046,469 uAUGs in the 216 multicellular plants and animals, although the number varied greatly across species.

The vast majority (>97%) of the uAUGs identified in the 478 eukaryotic species were start codons of putative canonical uORFs. Specifically, in a species, the percentage (mean ± s.e.) of nORFs, oORFs, and NTEs was 83.45 ± 0.41%, 14.24 ± 0.34%, and 2.31 ± 0.15%, respectively. The detailed information for the uORFs (nORFs and oORFs) and NTEs is presented in Supplementary Data 1.

### Purifying selection is the major force shaping the prevalence of uAUGs in eukaryotic genomes.

The number of uAUGs varied

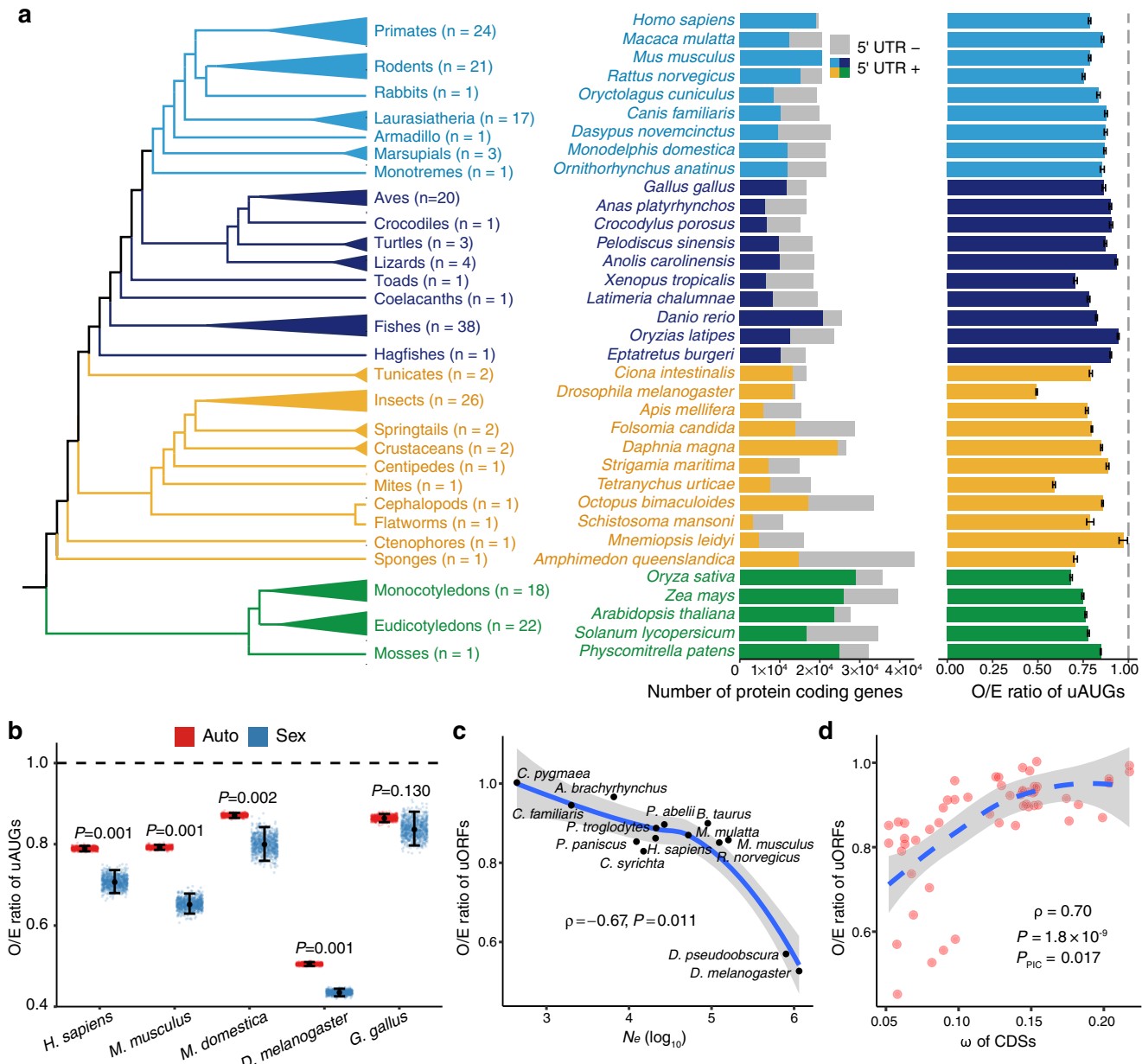

**Fig. 1 Variability of upstream AUG (uAUG) prevalence among eukaryotes and evolutionary driving forces. a** Overview of the 216 eukaryotes analyzed in this study. The left panel is the cladogram of the 216 eukaryotes. The number of species in each clade is shown in brackets. The middle panel shows the total number of protein-coding genes in 35 representative species. Genes with an annotated 5′ untranslated regions (5′ UTR+) are colored by clade, and those without 5′ UTR annotation (5′ UTR-) are shown in gray. The unavailability of annotated 5′ UTRs for many genes in less-studied organisms is presumably caused by the lack of accurate annotations. The right panel shows the ratio of the observed number of uAUGs to the expected number of uAUGs (O/E ratio) in the 35 species. The error bars indicate the 95% confidence interval of the O/E ratio. **b** O/E ratios of uAUGs in sex chromosome (X or Z) genes (Sex, blue) and autosomal genes (Auto, red) in humans, mice, opossum, flies, and chickens. $n = 1000$ permutation replicates for each category of genes in each species. Center point, median; error bars, 95% confidence intervals. $P$ values were obtained by two-sided Wilcoxon signed-rank tests, and no correction for multiple testing was made. **c** Relationship between the effective population size ($N_e$) and the O/E ratio of uORFs among 14 animals. The blue line indicates the local polynomial regression fit of the O/E ratio against $N_e$, and the gray band indicates the standard error of the fit. Spearman's correlation ($\rho$) between $N_e$ and the O/E ratio and the two-sided $P$ value are shown in the plot. **d** Relationship between the genome-wide median number of nonsynonymous changes per nonsynonymous site over the number of synonymous changes per synonymous site ($\omega$) of coding sequences (CDSs) and the O/E ratio of uORFs among 56 animals. The blue line indicates the local polynomial regression fit and the gray band indicates the standard error of the fit. Both Spearman's correlation and the significance of the two-sided phylogenetic independent contrast (PIC) between $\omega$ and the O/E ratio ($P_{PIC}$) are shown. Source data are provided as a Source Data file.

wildly across species, either due to the differences in the sequencing coverage of genomes, the accuracy and completeness of 5′ UTR annotation, the number of protein-coding genes, the length of 5′ UTRs, or mutational bias in 5′ UTRs[77]. To control for the compounding factors, in each species, we compared the observed number of uAUGs (O) versus the expected number (E) that was obtained with the assumption of randomness by randomly shuffling the 5′ UTR sequences. We maintained the same dinucleotide frequencies in each sequence during shuffling for two reasons. First, the stacking energy of a new base pair is influenced by the neighboring base pairs in an RNA molecule[78,79]. Second, the biased mutations in certain dinucleotide contexts, such as from CpG to TpG mutations in mammals, might also affect the occurrences of uAUGs. The O/E ratio enabled the efficient measurement of selective pressure on uAUG depletion in a given species. As expected[32,56–59], the O/E ratio of uAUGs was significantly lower than 1 in nearly all the examined species (473 out of 478 species, Fig. 1a and Supplementary Data 1). As a negative control, we also calculated the O/E ratio of all the other 63 possible triplets in 5′ UTRs and 3′ UTRs separately in each species. Of note, AUG had the lowest relative O/E ratio (5′ UTRs over 3′ UTRs) among all the 64 possible triplets (Supplementary Fig. S2), supporting the notion that purifying selection is the major force shaping the prevalence of uAUGs in the eukaryotic genomes. Interestingly, some AUG-like triplets (e.g., AUU, UUG, AUC, and GUG) tended to have higher O/E ratios in 5′ UTRs than in 3′ UTRs in all the clades. Such AUG-like triplets were either selectively maintained in 5′ UTRs as they can be used as noncanonical start codons, or alternatively, were the consequence of the depletion of uAUGs because point mutations can easily convert AUG to AUG-like triplets (e.g., from AUG → UUG) in the 5′ UTRs. However, further studies are required to separate these two possibilities.

Within a species, the O/E ratio of uAUGs was significantly lower in the 5′ UTR regions within a distance $L$ from the start codons of CDSs (cAUGs) than in the remaining 5′ UTR regions ($P = 3.5 \times 10^{-37}$, two-sided Wilcoxon signed-rank test when $L$ was set to 100 nt; other values of $L$ did not affect the conclusion, see Supplementary Fig. S3). This pattern is consistent with previous observations that uAUGs closer to CDSs showed a higher tendency to be depleted from 5′ UTRs[57]. Notably, the O/E ratio of oORFs was significantly lower than that of nORFs (Supplementary Fig. S4), suggesting oORFs tend to be more repressive and thus under stronger purifying selection than nORFs. Interestingly, NTEs showed lower O/E ratios than both oORFs and nORFs in 457 out of 478 species (Supplementary Fig. S4), suggesting that novel NTEs were selected against as they might alter protein functions[21].

X-linked mutations experience stronger selection than autosomal mutations if the fitness effects of the mutations are (partially) recessive[80–82]. If purifying selection is the dominant force acting on the occurrences of uAUGs in a genome, we expect to observe lower O/E ratios of uAUGs on X chromosomes than on autosomes. Indeed, significantly lower O/E ratios of uAUGs were found in X chromosomes than in autosomes, and this finding was obtained with both vertebrates and insects (Fig. 1b). In birds, which present female heterogamety (males ZZ, females ZW), selection is more efficient on the Z chromosome than autosomes[83]. Accordingly, a slightly lower O/E ratio of uAUGs was observed on the Z chromosome than on autosomes (Fig. 1b). Thus, the comparison between sex chromosomes and autosomes reinforces the thesis that purifying selection is the major force governing the prevalence of uORFs and NTEs in eukaryotes.

Overall, these results suggest that uAUGs were selected against in 5′ UTRs, and the NTEs, which only accounted for a small fraction (~2.31% on average) of the uAUGs, were also shaped by

strong purifying selection during evolution. Since uORFs (nORFs and oORFs) and NTEs might have different mechanisms in regulating gene expression and function, in what follows, we only focused on the putative canonical uORFs.

**Gene expression level as an important factor influencing the genome-wide distributions of uORFs across genes.** In humans, genes with uORFs exhibited lower expression levels than genes without uORFs[84]. Similarly, our analysis of previously published mRNA and protein abundance data of fly, human, mouse, mustard plant, and yeast revealed uORFs were infrequently detected in housekeeping genes, and there were significant anticorrelations between the gene expression level and the number of uORFs (Supplementary Fig. S5a and Supplementary Data 2). Meanwhile, gene ontology analysis revealed that genes containing putative uORFs tend to be enriched in the categories of signal transduction, transcription factors, and membrane proteins (Supplementary Fig. S5b; Supplementary Data 3). These patterns still held when we focused on the uORFs supported by previously published ribosome profiling data in fly[32] and other species collected in the GWIPs-viz database[85] (Supplementary Table S1; Supplementary Fig. S5b). Noteworthy, the anticorrelation between uORF occurrences and gene expression level well reconciles with the gene ontology analyses as housekeeping genes tend to be highly (or broadly) expressed[86].

Since gene expression level affects the efficacy of natural selection[87,88], we further asked whether the efficacy of purifying selection is reduced in removing deleterious uORFs in lowly expressed genes. We grouped genes of a species into 20 equal-sized bins based on increasing expression levels and calculated the O/E ratio of uORFs in each bin. In all the five species we examined, the O/E ratio was lower than 1 in each bin (Supplementary Fig. S5c), suggesting that purifying selection was the dominant evolutionary force acting on the uORF occurrence regardless of gene expression levels. Interestingly, we observed significant anticorrelations between the gene expression level and O/E ratio of uORFs in each species, suggesting that purifying selection acting on uORFs is relatively weak for lowly expressed mRNAs.

Thus, our results suggest that gene expression level is an important factor influencing uORF distribution across genes in a eukaryotic species. Excessive uORFs in highly expressed genes might cause insufficient protein output, which is harmful to the organisms. We postulate that purifying selection has removed deleterious uORFs in the highly expressed genes more efficiently than in the lowly expressed genes. On the other hand, genes in specific functional categories, such as transcriptional factors, which are likely to be lowly expressed, might be preferentially suppressed by uORFs at the translational level for optimizing protein production. Further studies are needed to investigate the relative importance of the two mechanisms in shaping the anticorrelation between gene expression level and uORF occurrence.

**Differences in $N_e$ influence interspecies differences in uORF occurrences.** The O/E ratio of uORFs varied widely across the eukaryotic species (Fig. 1a and Supplementary Data 1), suggesting that the efficacy of natural selection differs across these species. Because the efficiency of natural selection is determined by $N_e$[89], we questioned whether the differences in the O/E ratios of uORFs between different eukaryotes are due to the differences in $N_e$. We reasoned that the O/E ratio should be lower in species with a larger $N_e$ because purifying selection is the dominant force acting on uORFs, and deleterious uORFs will be depleted more efficiently by purifying selection. Indeed, we uncovered a significant

negative correlation between the O/E ratio and $N_e$ (Spearman's ρ $=-0.67$, $P = 0.011$) for 14 animals for which the $N_e$ value was estimated in previous studies (Fig. 1c and Supplementary Table S2).

Because the $N_e$ value was unknown for most eukaryotes investigated in this study, we calculated the genome-wide average $dN/dS$ ratio (ω, number of nonsynonymous changes per nonsynonymous site over the number of synonymous changes per synonymous site) of CDSs between closely related species as an indirect measure of the average $N_e$ for a clade based on the following rationale: if a clade includes species with a large $N_e$, deleterious nonsynonymous mutations in CDSs will be more efficiently removed by natural selection, resulting in a smaller ω value for that clade. Therefore, if the purifying selection is the major force acting on uORF prevalence, a positive correlation between the O/E of uORFs and the ω of CDSs would be expected across different species. We aligned orthologous CDS sequences at the genomic scale for 37 pairs of closely related species, and calculated the genome-wide ω value for each pair of species (Supplementary Table S3). In this analysis, we assumed that two closely related species would have the same ω values and obtained both the O/E ratios of uORFs and the ω values of CDSs for 56 species. We uncovered a significant positive correlation between the O/E ratio and the ω value (ρ = 0.70, $P = 1.8 \times 10^{-9}$; Fig. 1d), which further confirms that the differences in $N_e$ determine the differences in uORF depletion among eukaryotic genomes. Interestingly, a significant positive correlation between the median 5′ UTR length and the ω was also observed (ρ = 0.54, $P = 1.4 \times 10^{-5}$; Supplementary Fig. S6), suggesting that the 5′ UTR length is also under selective constraints. This finding is not surprising because the number of uORFs is generally positively correlated with the 5′ UTR length[90]. To exclude the possibility that the observed positive correlations were confounded by the phylogenetic relationships of the eukaryotic species, we also performed phylogenetic independent contrasts[91] and still detected significant positive correlations between the O/E ratio and the ω ($P = 0.017$) and between the 5′ UTR length and the ω ($P = 0.021$). Together, our analyses suggest that purifying selection is the dominant force governing the contents of uORFs in eukaryotes and that the degree of uORF depletion in a species is mainly determined by the $N_e$ of that species.

**Role of positive selection in influencing the prevalence of uORFs in eukaryotes.** Although uAUGs are generally depleted in the 5′ UTRs of *Drosophila*, our previous results indicated that a considerable fraction of the uORFs recently fixed in *D. melanogaster* were driven by positive Darwinian selection[32]. Our results are consistent with the notion that the very large $N_e$ of *D. melanogaster* increases the efficacy of both positive selection and purifying selection[89]. Nevertheless, whether positive selection drives the fixation of uORFs in a eukaryote with a small $N_e$, such as humans, remains unclear. To address this research gap, we analyzed the new uORFs that were newly fixed in the lineages leading to extant humans after divergence from *Pongo abelii*, *Gorilla gorilla*, or *Pan troglodytes* using the asymptotic McDonald-Kreitman test (asymptoticMK)[92,93]. We detected weak signals of positive selection on the newly fixed uORFs in all three branches, and the value of $\alpha_{asym}$, which represents the fraction of newly formed uORFs driven to fixation by positive selection, was 0.24 (95% confidence interval [CI], −0.04–0.51), 0.20 (95% CI, −0.09 to 0.49), and 0.19 (95% CI, −0.10 to 0.48) in the three branches, respectively (Fig. 2a). Noteworthy, C>T mutations at CpG dinucleotides are highly frequent in mammals[94], and new AUGs can be generated from CpG to TpG mutations through two approaches[95]: (1) from ACG to ATG, and

(2) from CGTG to CATG (Fig. 2b). Thus, we further examined new uORFs derived from the CpG contexts and the remaining new uORFs separately. Roughly speaking, ~33% of the new uORFs fixed in each of the three branches were generated by CpG to TpG mutations. Interestingly, the CpG-derived uORFs were under strong positive selection (the $\alpha_{asym}$ was 0.48 (95% CI, 0.18–0.78), 0.45 (95% CI, 0.13–0.76), and 0.44 (95% CI, 0.11~0.76) in the three branches, respectively), while the $\alpha_{asym}$ for the remaining uORFs was close to 0 (Fig. 2b). Noteworthy, the $\alpha_{asym}$ values were even higher when we focused on the new uORFs that were derived from the CpG contexts in the highly expressed genes (Supplementary Table S4). Of note, for the new uORFs fixed in *D. melanogaster* we previously analyzed[32], a higher $\alpha_{asym}$ value was also observed for the highly expressed genes (Supplementary Table S4). Therefore, although the prevalence of uORFs in a species was generally under purifying selection, we still found a fraction of uORFs might be favored by positive selection even in primates that typically have a small $N_e$.

To further explore how positive and purifying selection coupled with differences in $N_e$ shaped the repertoire of uORFs in a given species, we mathematically modeled the O/E ratio of uORFs by treating this ratio as the average fixation probability of mutations with different fitness effects. Considering that uORFs are generally deleterious, we assumed that 20%, 75%, and 5% of newly originated uORFs are neutral, deleterious, and beneficial, respectively. We also assumed that both beneficial and deleterious mutations present the same absolute selection coefficient in a diploid organism and that the fitness reduction in heterozygotes is half of that in homozygous mutants. We then calculated the overall fixation probability of newly originated uORFs relative to the neutral expectation, which is, by definition, similar to the O/E ratio. In our modeling, the relative fixation probability of newly originated uORFs gradually decreased with increases in the $N_e$ (Fig. 2c), which resembled our observation that species with larger $N_e$ values tend to exhibit lower O/E ratios. Moreover, a higher fraction of fixed uORFs that are driven by positive selection was obtained with a higher $N_e$ value (Fig. 2d). Together, our results suggest that both purifying selection and positive selection act on uORF occurrences during eukaryotic evolution and that differences in $N_e$, which affects the efficiency of both types of natural selection, plays a major role in shaping the differences in uORF prevalence among eukaryotic species.

**Selective constraints on start codons of uORFs in eukaryotes.** Next, we questioned how the uORFs were maintained during eukaryotic evolution. The start codon is the most important definitive characteristic of a uORF[4], and a uORF with a more conserved start codon tends to be more repressive toward the translation of the downstream CDS[19,32]. Here, we first quantitatively measured the selective pressures on the AUG start codons of uORFs (uoAUGs) in vertebrates, insects, and yeasts. Among the 78,003 uoAUGs identified in the human reference genome, 98.7% have conserved uoAUGs in other vertebrates (Fig. 3a). Interestingly, 1030 (1.3%) uoAUGs were only observed in humans and not in any other species, suggesting that these have a recent origin. Whether the human-specific uoAUGs are associated with unique human features remains to be investigated. For each uoAUG identified in the human reference genome, we calculated the branch length score (BLS) based on the conservation patterns of the orthologous sites among 100 vertebrate species using a previously described method[96] (Fig. 3b). To estimate the number of uoAUGs that are more conserved than the neutral expectation, we also calculated the BLS for all 63 other triplets present in 5′ UTRs based on the assumption that these triplets evolve neutrally. The start codons of 173,290 noncanonical

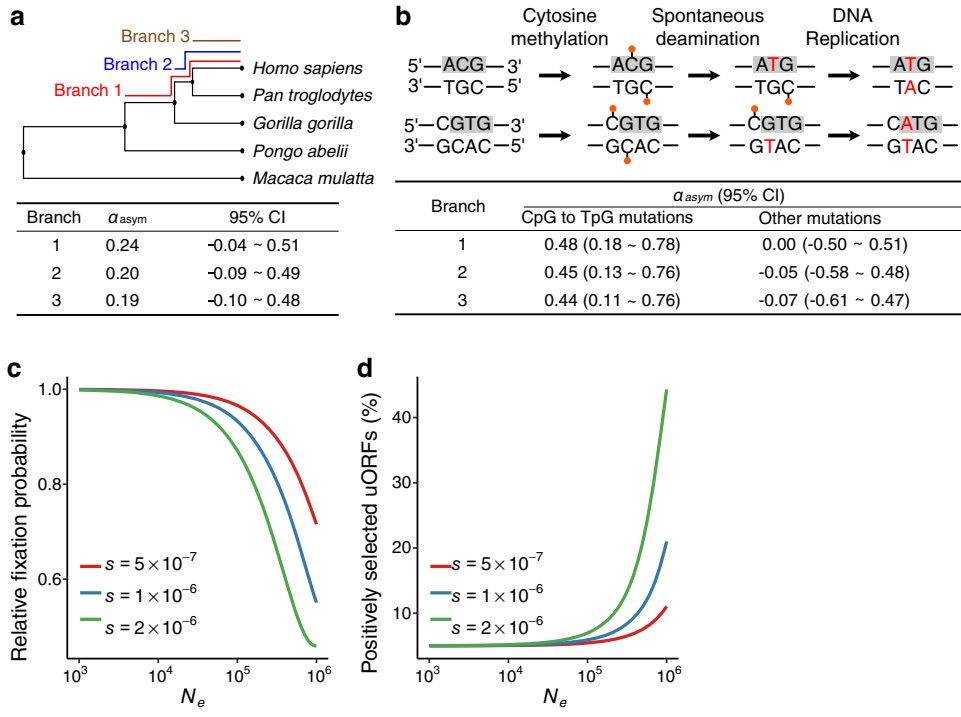

**Fig. 2 Selection and effective population size ($N_e$) shape the upstream open reading frame (uORF) prevalence in eukaryotes. a** Asymptotic McDonald–Kreitman (AsymptoticMK) test of newly fixed uORFs in the lineages leading to extant humans (branches 1, 2, and 3). The left panel shows the phylogeny of the five primates related to the analysis. Rhesus macaque (*Macaca mulatta*) was used as the outgroup. The fraction of newly fixed uORFs driven by positive selection ($\alpha_{asym}$) is shown in the bottom panel. **b** The result of AsymptoticMK tests for newly fixed uORFs derived from CpG to TpG mutations and the other mutations on each branch. The two approaches by which CpG to TpG mutations create new ATGs in 5′ UTRs are illustrated above. Relative fixation probability of newly originated uORFs (**c**) and the fraction of uORFs driven by positive selection (**d**) as a function of the $N_e$ of a simulated population. In the simulation, we assumed that beneficial and deleterious mutations presented the same absolute selective coefficient (*s*) and that there was no dominance ($h = 0.5$). The fractions of newly originated uORFs that are deleterious, neutral, or beneficial are 75%, 20%, and 5%, respectively. Source data are provided as a Source Data file.

uORFs identified in humans by McGillivray et al.[17] were excluded from the neutral controls.

Compared with the other triplets, uoAUGs showed significantly higher BLS values ($P = 7.6 \times 10^{-58}$, two-sided Wilcoxon rank-sum test [WRST]; Fig. 3c), suggesting that the uoAUGs are under selective constraints during evolution. At a BLS cutoff of 0.5, the signal-to-noise ratio (fraction of uoAUGs that meet a minimum BLS cutoff divided by the fraction of other triplets with the same minimum BLS) was 3.71, and this value increased with increases in the BLS cutoff (Fig. 3d). Moreover, the BLS values of translated uoAUGs supported by ribosome profiling data from human samples were significantly larger than those of untranslated uoAUGs ($P = 8.6 \times 10^{-262}$, two-sided WRST; Fig. 3c). Accordingly, at a BLS cutoff of 0.5, a markedly higher signal-to-noise ratio (4.40) was obtained for the translated uoAUGs (Fig. 3d), suggesting that uoAUGs from which translation is initiated are under even stronger functional constraints. We also calculated the BLS values for the start codons of the 173,290 noncanonical uORFs previously identified in humans by McGillivary et al.[17]. Since conservation was used as a feature to identify the noncanonical uORFs in that study, it is not surprising that these noncanonical start codons were slightly (~1.2 times) more conserved than the other random triplets ($P = 2.1 \times 10^{-77}$, two-sided WRST; Fig. 3d). However, they were significantly less conserved than the canonical uoAUGs ($P = 1.3 \times 10^{-12}$, two-sided WRST).

The uoAUGs identified in *D. melanogaster* were also more conserved than the random triplets in 5′ UTRs across the 27 examined insect species (Fig. 3e, f and Supplementary Fig. S7),

and the uoAUGs with translational evidence from ribosome profiling data were more conserved (Figs. 3e, f). Analogously, the uoAUGs in *S. cerevisiae* were significantly more conserved than the other triplets in the 5′ UTRs across the seven yeast species we examined, no matter we used all the uoAUGs or the translated ones only ($P < 9.5 \times 10^{-11}$, two-sided WRST; Supplementary Fig. S8). Altogether, the start codons of the canonical uORFs, particularly the translated ones, are more likely to be maintained by functional constraints during eukaryotic evolution.

**Coding regions of uORFs are overall under neutral evolution.** How many uORFs can encode functional peptides remains unclear[4,10,32,97]. If a uORF encodes a functional peptide, one expects that the coding region of that uORF should be under selective constraints. In contrast, if the function of a uORF is to tune the translation of the downstream CDS by sequestering or competing for ribosomes, the coding regions of uORFs might be under neutral evolution or weaker selective constraints. Thus, we investigated the conservation patterns of the uORF peptides in the vertebrates, insects, and yeasts. While NTEs were not included as uORFs in our analysis, we further excluded CDS-overlapping portions of oORFs due to the confounding effects of selective constraints on CDS evolution. Briefly, for each of the 48,286 human uORFs that encode peptides of at least ten amino acids, we searched the putative homologous peptide sequences in other vertebrate species and calculated the BLS for that peptide (homologous sequences that have stop codons or frameshifts within 80% of the start regions were excluded). 48.8% of these

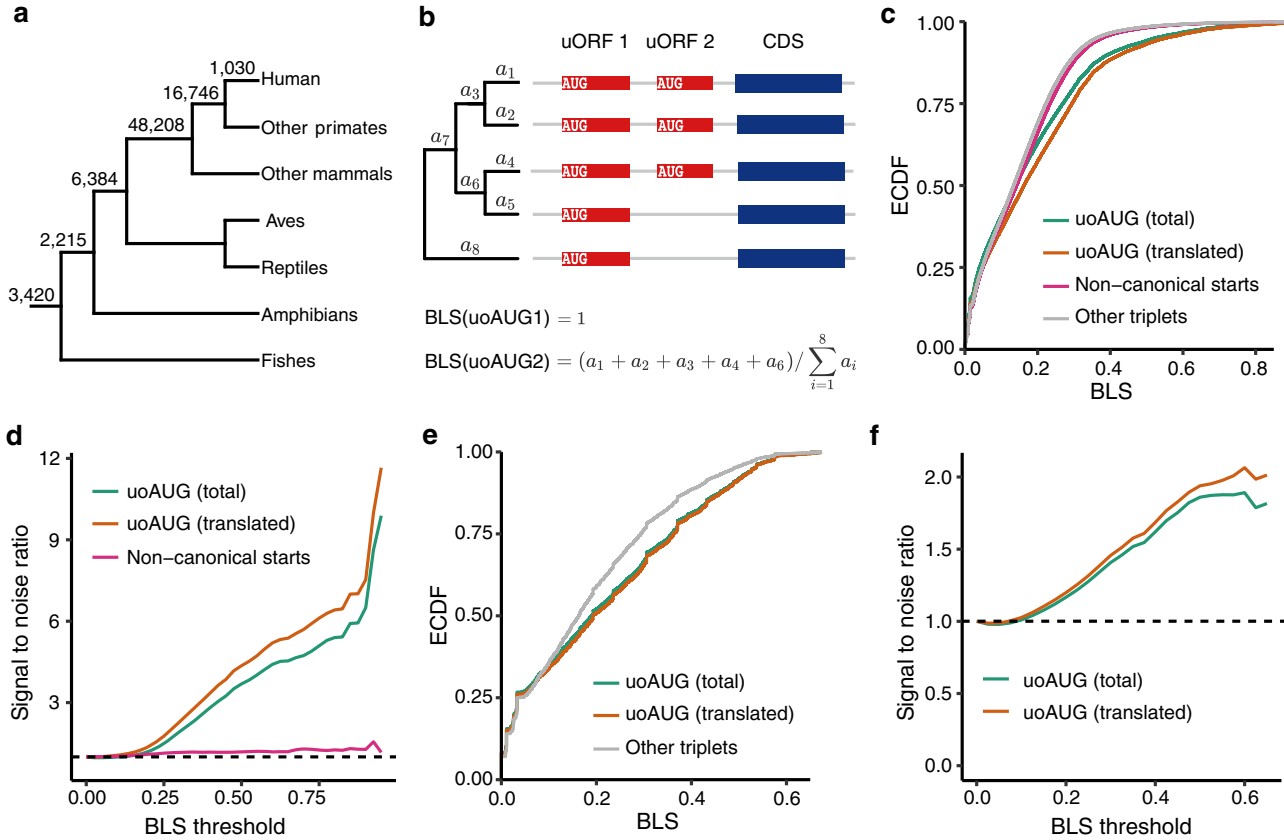

**Fig. 3 Selective constraints on the upstream open reading frame (uORF) start codons. a** Age distribution of start codons (uoAUGs) of human uORFs. The number of origination events assigned to each branch was inferred with the maximum parsimony method. **b** The scheme showing how the branch length scores (BLSs) for the start codons of two uORFs are calculated based on their presence or absence across species. In this hypothetical example, the length of each branch is denoted with $a_i$ ($i = 1$–8). **c** Empirical cumulative distribution function (ECDF) of the BLSs for uoAUGs, noncanonical start codons, and the other triplets in human 5′ untranslated regions (UTRs). The BLS of uoAUGs (total or translated) or noncanonical start codons was significantly larger than that of the other triplets ($P < 8 \times 10^{-58}$, two-sided Wilcoxon rank-sum tests). **d** Signal-to-noise ratios of the BLSs of uoAUGs and noncanonical start codons relative to other triplets in humans based on different thresholds of minimum BLS. The dashed line delineates a signal-to-noise ratio of 1 expected under neutral evolution. **e** ECDF of BLS for uoAUGs and the other triplets in fly 5′ UTRs. The BLS of uoAUGs (total or translated) was significantly larger than that of the other triplets ($P < 2 \times 10^{-88}$, two-sided Wilcoxon rank-sum tests). **f** The signal-to-noise ratio of BLSs for uoAUGs relative to other triplets in flies based on different minimum BLS thresholds. Source data are provided as a Source Data file.

human uORFs putatively presented conserved peptide sequences only in primates; 36.6% of them putatively exhibited conserved peptide sequences in mammals other than primates, and 1.82% of them putatively exhibited conserved peptide sequences in fishes. Of note, the BLS values of the uORF coding sequences were significantly lower than those of the uoAUGs (Fig. 4a; Supplementary Fig. S9 for other cutoffs of the minimum number of AAs required for uORF peptides). Analogously, for the uORFs identified in *D. melanogaster*, the coding regions of uORFs were also less conserved than uoAUGs in the 27 examined insect species (Fig. 4b and Supplementary Fig. S9). A similar pattern was also observed in the seven-way alignments of yeasts (Supplementary Fig. S10). Of note, a strong anticorrelation was observed between the BLSs and the lengths of uORF peptides in both humans and flies (see Fig. 4c and d), suggesting the peptides encoded by long uORFs are less likely to be maintained during evolution because they were more likely disrupted by stop codons or frameshifts. Also, if the major function of uORFs is to regulate CDS translation, a longer uORF might be less advantageous than a shorter one because the translation of a longer uORF consumes more energy and metabolites, which might be harmful to the host organisms. (The analysis was not conducted in yeasts because for 69% of the uORF peptides in *S. cerevisiae* we could not reliably identify the orthologous sequences in other yeast species).

Together, these results suggest that the coding regions of uORFs tend to be less conserved than start codons of uORFs.

To further test the selective pressure on the coding sequences of uORFs, we calculated the $\omega$ for coding regions of uORFs between humans and macaques. To reduce the noise in estimating $\omega$ values, we ranked the uORFs based on the Kozak scores of their start codons and equally grouped the uORFs into 1000 bins. For each bin, we concatenated the alignments of the uORF coding sequences and calculated the $\omega$ value. In contrast to CDSs, which present $\omega$ values markedly lower than 1, the $\omega$ value of the uORF coding region was roughly equal to 1 between humans and macaques (median $\omega = 1.05$; Fig. 5a and Supplementary Fig. S11a). Similarly, the $\omega$ of uORFs was also close to 1 between *D. melanogaster* and *D. simulans* (median $\omega = 0.99$ for all uORFs or 0.98 for translated uORFs only; Fig. 5b and Supplementary Fig. S11b). Moreover, we also grouped the single nucleotide polymorphisms (SNPs) in uORFs of humans (1000 Genomes Project[98]) and flies (*Drosophila* Genetic Reference Panel[99]) based on the derived allele frequencies (DAF) and calculated the ratio of nonsynonymous SNPs to synonymous SNPs ($pN/pS$) in each bin. In parallel, we performed the same analyses on SNPs in CDSs. In CDSs of both humans and flies, the $pN/pS$ ratios were substantially lower than the values expected under randomness, and the $pN/pS$ ratio was significantly

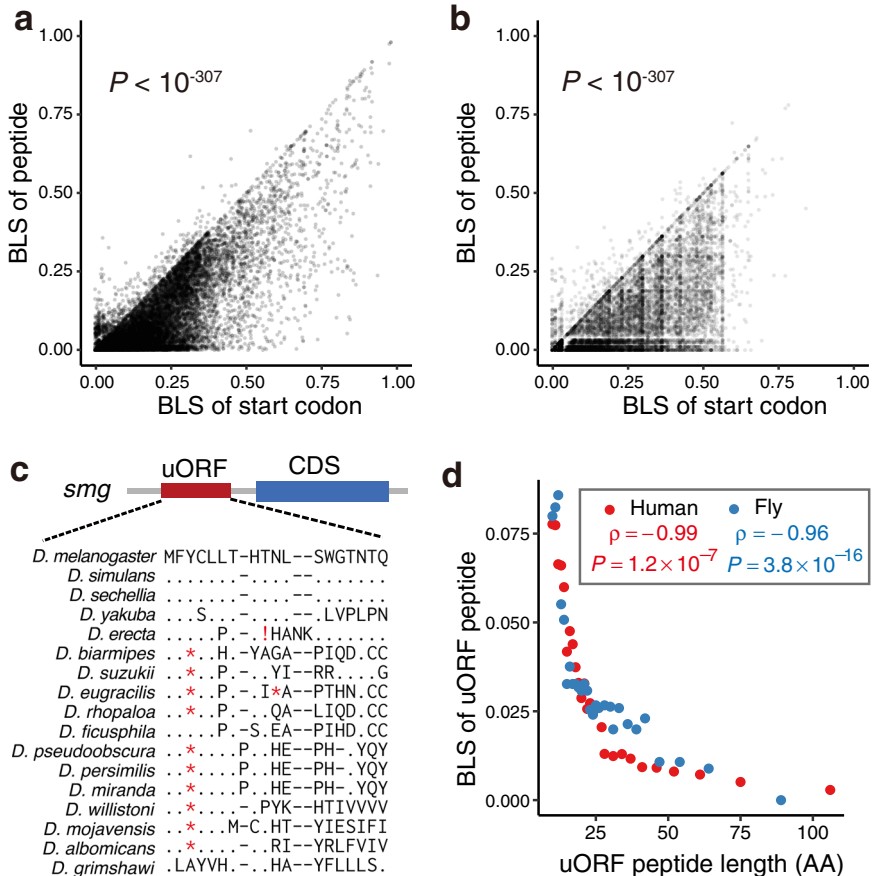

**Fig. 4 Conservation of upstream open reading frame (uORF)-encoded peptides.** The branch length score (BLS) of the coding region of a uORF is significantly lower than that of the start codon of that uORF in humans (**a**) and flies (**b**). **c** Example of a typical uORF with a conserved start codon in the fly *smg* gene. The orthologous peptide sequences in distant lineages exhibit many nonsynonymous substitutions and are frequently disrupted by stop codons (*) or frameshifts (!). **d** Relationship between the length and the BLS of uORF peptides in humans and flies. The uORFs were grouped into custom bins of increasing peptide length. The median peptide length and BLS value of each bin are displayed and were used to calculate Spearman's correlations ($\rho$) with two-sided $P$ values. "AA" refers to amino acid. Source data are provided as a Source Data file.

negatively correlated with the DAF bins in both species (Fig. 5c, d; Supplementary Fig. S11c, d). In contrast, in uORFs of both humans and flies, the *pN/pS* ratio fluctuated around expected values, and there was no correlation between *pN/pS* and DAF bins. Thus, these contrasting patterns indicated that at the population level, the nonsynonymous SNPs in CDSs were under strong purifying selection, while the nonsynonymous SNPs in uORFs were nearly neutral. Together, these analyses further revealed that the coding regions of uORFs are overall under neutral evolution in both primates and flies.

To estimate the proportion of uORFs that might encode conserved peptides, for each uORF, we also calculated PhyloCSF score, which predicts whether a genomic region potentially represents a conserved protein-coding region or not based on multiple sequence alignments[100] (a positive PhyloCSF score means that region is more likely to encode a peptide). As a negative control, we also calculated the PhyloCSF scores for 20,000 randomly selected ORFs in 3′ UTRs (downstream ORFs, dORFs), as these dORFs have little chance of translation. Among the 36,655 uORFs that are ≥10 codons and evidenced of translation in humans, only 361 (0.985%) had positive PhyloCSF scores (Supplementary Fig. S12a). In contrast, the PhyloCSF score was positive for 0.545% (109 out of 20,000) dORFs. Thus, after controlling for the background noises, only 0.44% (161) of the translated uORFs showed evidence of encoding conserved peptides. In *Drosophila*, 1.19% (152 of 12,745) translated uORFs

and 0.39% (78 out of 20000 dORFs) had positive PhyloCSF scores (Supplementary Fig. S12b), yielding an estimate of 0.80% (102 of 12,754) of the translated uORFs might encode conserved peptides. Overall, these analyses suggest that <1% canonical uORFs might encode conserved peptides.

To test whether our evolutionary analyses of uORFs were supported by experimental evidence, we analyzed the mass spectrometry (MS) data from 38 samples of different developmental stages or tissues of *D. melanogaster* (Supplementary Data 4)[101–105]. Among the 23,321 uORFs that met our parameter settings (Methods), 57 (0.24%) had peptides detected in at least one sample (Supplementary Data 5). Interestingly, the BLS analysis revealed that the MS-supported uORFs present slightly more conserved coding regions than the other uORFs (Fig. 5e), suggesting these MS-supported uORF peptides might be functionally important. Collectively, our results support the notion that most uORFs play regulatory roles and their start codons are maintained due to functional constraints, and only a tiny fraction (<1%) of the uORFs might encode peptides that are maintained by natural selection during evolution.

**Evolution of Kozak sequence contextual characteristics that influence uORF translation.** The Kozak sequence context (−6 to +4 nucleotides) around the uoAUG plays a prominent role in influencing the translational initiation of that uORF[16,32,106,107].

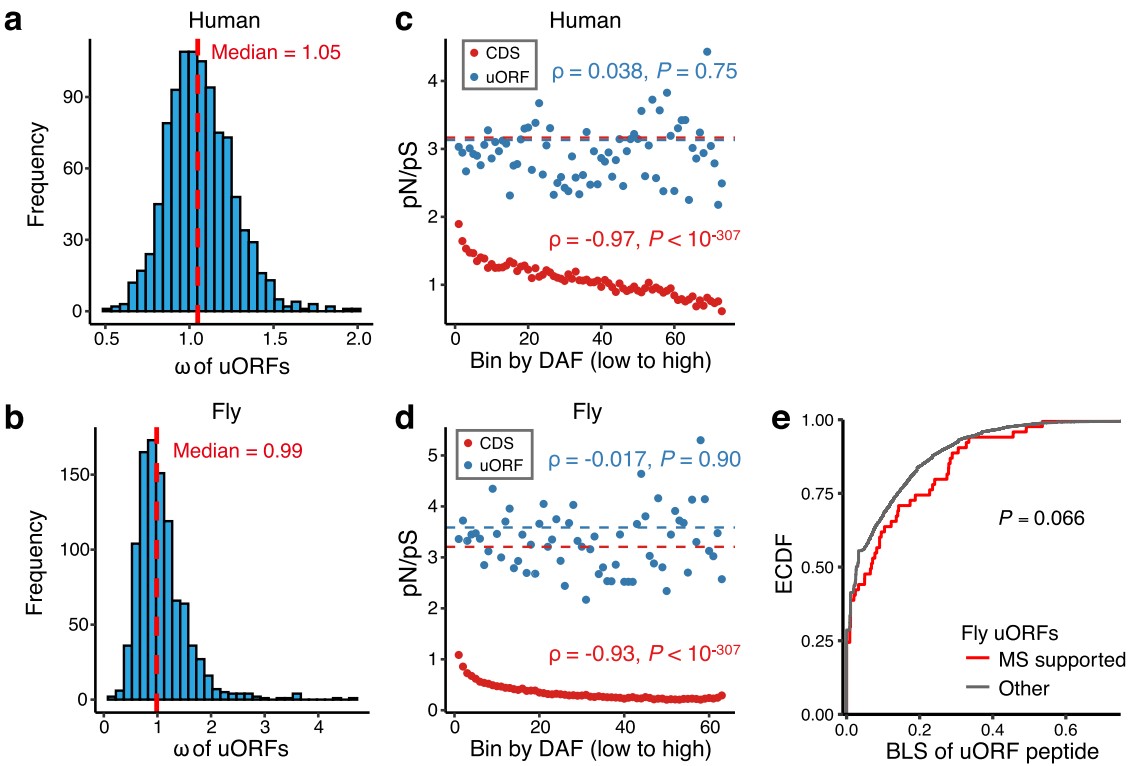

**Fig. 5 Selective constraints on coding regions of upstream open reading frames (uORFs). a** Distribution of the number of nonsynonymous changes per nonsynonymous site over the number of synonymous changes per synonymous site ($\omega$) of uORFs between humans and rhesus macaques. Human uORFs were equally divided into 1000 bins based on the start codon of uORFs with an increasing Kozak score. For each bin, the alignments of uORF sequences between human and rhesus macaque were concatenated to calculate the $\omega$ value. **b** Distribution of the $\omega$ values of uORFs between *D. melanogaster* and *D. simulans*. The procedure for $\omega$ calculation was similar to that described in **a**. **c** The ratio of the nonsynonymous to synonymous SNP numbers ($pN/pS$) in coding sequences (CDSs, red) and uORFs (blue) in bins with an increasing derived allele frequency (DAF). Spearman's correlation ($\rho$) between the $pN/pS$ ratio and the median DAF of each bin of uORFs and CDSs is displayed in the plot with two-sided *P* values. **d** Same as **c** but for fly uORFs. **e** The empirical cumulative distribution function of (ECDF) of peptide branch length score (BLS) for mass spectrometry (MS)-supported uORFs and the remaining uORFs in flies. uORFs with <10 amino acids were excluded. The one-sided *t*-test was performed to test differences in BLS. Source data are provided as a Source Data file.

As the nucleotide compositions differ between eukaryotes[108–110], the preferential Kozak sequence context around cAUGs also differs across species[111,112]. Nevertheless, whether Kozak contextual characteristics around uoAUGs evolve remains unclear. To address this research gap, in each of the 478 species, we reconstructed a position weight matrix of the Kozak sequence context (PWMK) for all the CDSs (Supplementary Data 6 and Supplementary Fig. S13). Subsequently, in each species, we calculated the Kozak score for each cAUG or uoAUG with the PWMK of that species as previously described[32].

To test the performance of the Kozak score in predicting the translational initiation of a uORF (or CDS), we analyzed translation initiation site (TIS) profiling data from three species (human, mouse, and fly)[32,38,40]. We detected the translation of 26,344, 16,245, and 15,195 canonical uORFs that were supported by TIS data in at least one sample for human, mouse, and fly, respectively (Supplementary Table S1). For each uoAUG in a sample, we calculated the normalized TIS signal by dividing its initiating ribosome-protected fragment (RPF) count by its mean coverage in the matched RNA-Seq data. Strong positive correlations were found between the Kozak score and the normalized TIS signal for both cAUGs (Supplementary Fig. S14) and uoAUGs (Fig. 6a and Supplementary Fig. S15), suggesting that start codons with an optimized Kozak sequence context exhibit a higher translation initiation efficiency for both CDSs and uORFs.

Interestingly, the number of uORFs was negatively correlated with the Kozak score of the cAUGs in most species (Fig. 6b), suggesting that uORFs tend to suppress genes translated at low levels, as previously suggested[59]. Also, the Kozak scores of the uORFs were significantly lower than those of the CDSs in each species (Supplementary Fig. S16a), supporting the notion that uORFs are generally located in less optimal contexts than CDSs[16,32,113]. To test whether the sequence contexts of uoAUGs are optimized, in each species, we also calculated the Kozak scores of the AUG triplets in 3′ UTRs (downstream AUGs, dAUGs) as neutral controls. The Kozak scores of uoAUGs were significantly higher than those of dAUGs in most (82.4%, 112 out of 136) vertebrates, (61.0%, 25 out of 41) plants, and (71.9%, 174 out of 242) fungi; however, an opposite trend was observed in invertebrates, and no obvious trend was observed in protists (Supplementary Fig. S16b). These results suggest that the optimization of the Kozak sequence context of uORFs is different across eukaryotic clades.

To examine whether the Kozak contextual characteristics of uORFs evolved, in each of the 478 species, we calculated the pairwise Euclidian distance of the PWMK for uORFs (or CDSs) between two species ("Methods"). Interestingly, for both uORFs and CDSs, the distance between two species from a clade tend to be significantly shorter than that between one species in that clade and another species outside of that clade (Fig. 6c). A similar pattern was observed for the uORF PWMK as well (Fig. 6d).

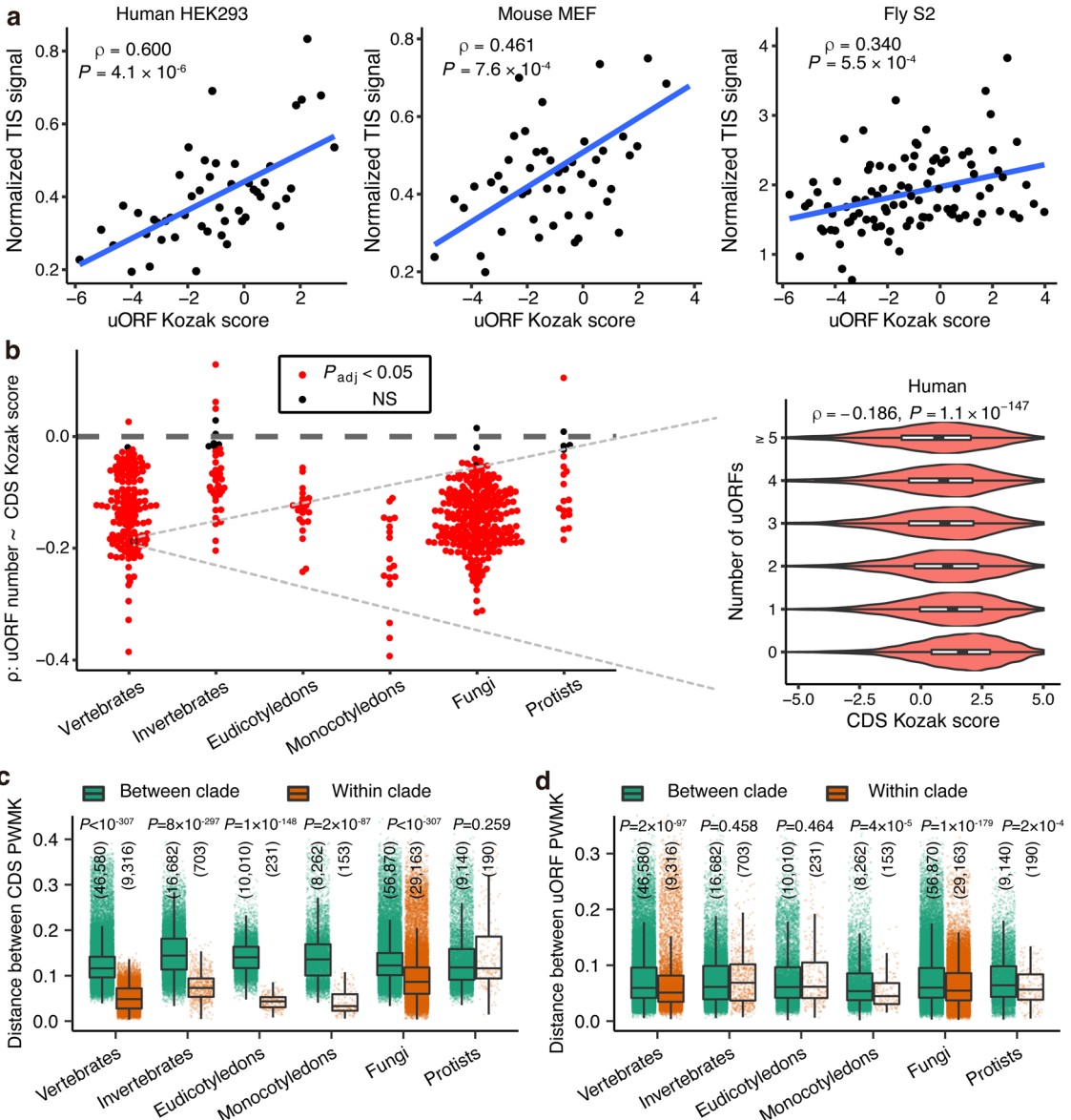

**Fig. 6 The Kozak sequence contextual characteristics that influence upstream open reading frame (uORF) translation. a** Relationship between the Kozak scores and normalized translational initiation signals of uORF start codons (uoAUGs) in human HEK293 cells, mouse MEF cells, and fly S2 cells. In each sample, we ranked uORFs based on increasing Kozak scores and divided them into 50 bins (100 bins for S2 cells) with equal numbers of uoAUGs. The median Kozak score and normalized TIS signal for each bin were used to calculate Spearman's correlations ($\rho$) and two-sided $P$ values. The linear fit was indicated with a blue line. **b** The distribution of Spearman's correlation coefficients between the coding sequence (CDS) Kozak scores and the number of uORFs for that gene in eukaryotes in different taxa. In the left panel, each dot represents one species. The right panel shows that in humans, genes that have multiple uORFs tend to have weaker Kozak sequence context around the start codon of CDSs. $P_{adj}$, two-sided $P$ value after correction for multiple testing; NS, not significant. Box plots showing the distribution of the Euclidian distance of the position weight matrix of Kozak sequences (PWMK) for cAUGs (**c**) and uoAUGs (**d**) between species within the same taxa (brown) or species in different taxa (green). Differences in distances were compared with two-sided Wilcoxon rank-sum tests. Exact $P$ values (no correction for multiple testing were made) and the number of pairwise distances in each group were shown in the plot. Center line, median; box limits, upper and lower quartiles; whiskers, 1.5 times the interquartile range. Source data are provided as a Source Data file.

Together, our results suggest that, although the Kozak sequence context plays a pivotal role in regulating the translational initiation of uORFs and CDSs in eukaryotes, its contextual characteristics evolved during eukaryotic evolution.

**Comparing the canonical versus noncanonical uORFs in repressing CDS translation in human populations.** Recent studies have demonstrated that noncanonical uORFs are very abundant[17,34], and many of them might have diverse

functions[4,20]. Moreover, hundreds of noncanonical uORFs are conserved between different yeast species, suggesting they might be functionally important[114]. In the above analyses, we mainly focused on the canonical uORFs because (1) the majority of the species analyzed in this study had no ribosome profiling data available, and (2) it is still challenging to identify noncanonical uORFs without experimental data.

To test whether the noncanonical uORFs influence the translation of CDSs, we extracted high-quality genotyping,

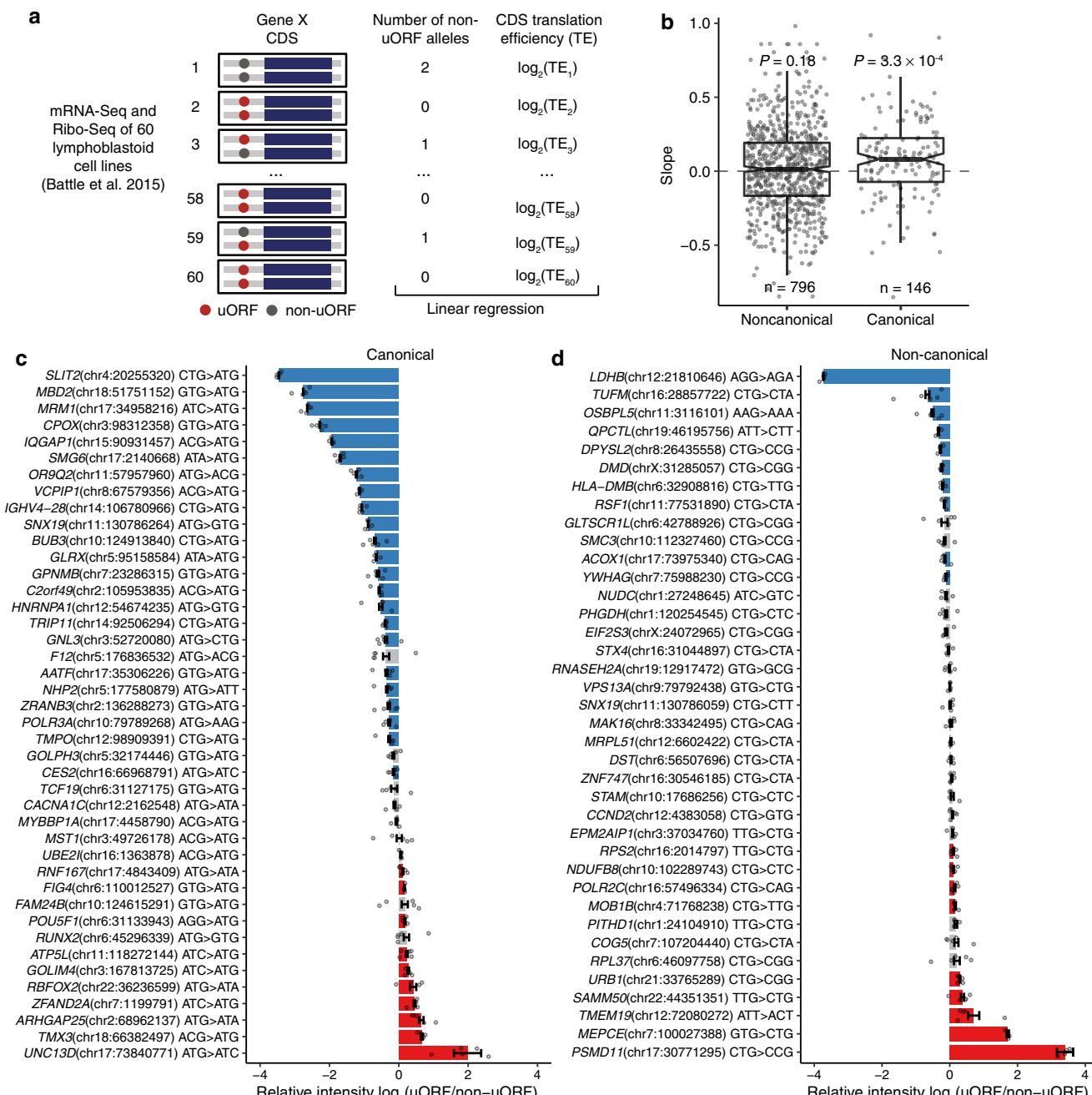

**Fig. 7 Experimental verification of canonical and noncanonical upstream open reading frames (uORFs). a** The scheme showing how to determine the effect of uORF variations in the human population on the translation efficiency (TE) of downstream coding sequences (CDSs). With the mRNA-Seq and Ribo-Seq data of 60 human lymphoblastoid cell lines[115], we calculated the translation efficiency of CDS for each gene and obtained the genotypes of each subject from the 1000Genomes Project. For each uORF variant, we performed a linear regression between the number of non-uORF alleles and the $\log_2(TE)$ of downstream CDS across the 60 cell lines. **b** The distribution of slopes in the linear regression between genotypes (the number of non-uORF alleles) and CDS translational efficiency among 60 cell lines. Center line, median; box limits, upper and lower quartiles; whiskers, 1.5 times the interquartile range. The number of variants in each category was shown in the plot. Exact $P$ values of two-sided Wilcoxon signed-rank tests were shown in the plot. The ratio of relative luciferase intensity ($\log_2$) between the reporters with the uORF allele or the non-uORF allele for each variant of canonical uORFs (**c**) or noncanonical uORFs (**d**). The bars are displayed in blue or red when the relative intensity of uORF-allele is significantly lower or higher than that of the non-uORF allele (one-sided Wilcoxon rank-sum tests, $P_{adj} < 0.1$), respectively. Measures of center, mean; error bars, standard errors. $n = 4$ or 5 independent biological replicates for each variant (details are presented in source data). Source data are provided as a Source Data file.

mRNA-Seq, and Ribo-Seq data of 60 human lymphoblastoid cell lines from previous studies[98,115], and examined whether variations in uORF start codons influence the translation efficiency of the main CDSs among different samples (Fig. 7a). Among the potentially functional uORFs in humans predicted by McGillivray et al.[17], 146 canonical and 796 noncanonical uORFs had genetic

variants in their start codons among these samples (only variants with minor allele frequency ≥5% were considered in the analysis). We performed linear regressions to assess the regulatory impact of uORF alteration on the translation of down-stream CDSs, with a positive slope value in the regression meaning that the presence of a uORF in certain individuals is associated with a decrease in

the translation efficiency of the downstream CDS in those individuals, and vice versa ("Methods"). A general trend was the slope values were overall positive for the canonical uORFs, while the slope values for the noncanonical uORFs fluctuated around 0 (Fig. 7b). This comparison suggests that in human populations, the noncanonical uORFs overall have relatively limited repressive effects on CDS translation compared to the canonical uORFs, although we cannot exclude the possibility that a small fraction of the noncanonical uORFs might have strong repressive effects on the translation of downstream CDSs.

To experimentally verify the influence of both types of uORFs on CDS translation, we sampled 80 human uORFs and performed luciferase reporter assays in HEK293FT cells (Supplementary Fig. S17). These tested uORFs, which included 42 canonical and 38 noncanonical ones, were predicted potentially functional by McGillivray et al.[17] and had polymorphic start codons in human populations. For each uORF, we compared the repressive effect of the annotated uORF allele versus that of the non-uORF allele in suppressing translation of the reporter gene. Although occasionally the non-uORF allele had a stronger repressive effect than the uORF allele, the general trend was that the uORF allele had a stronger effect than the non-uORF allele in suppressing translation (Fig. 7c, d). Moreover, a significantly higher proportion of the canonical (55%, 23/42) than the noncanonical (26%, 10/38) uORFs exhibited the pattern that the annotated uORF allele showed a significantly stronger repressive effect on the CDS translation than the non-uORF allele ($P = 0.013$, Fisher's exact test, Fig. 7c, d). Also, the difference in CDS translation suppression between the uORF and the non-uORF allele is significantly larger for the canonical than the noncanonical uORFs ($P = 0.006$, one-sided WRST). Altogether, these results reinforced the thesis that the noncanonical uORFs overall have weaker repressive effects on CDS translation than the canonical uORFs.

## Discussion

In this study, we analyzed ~17 million uAUGs, $97.69 \pm 0.15\%$ of which are start codons of putative canonical uORFs in 478 eukaryotic species that span the majority of extant taxa of eukaryotes. Although the prevalence of canonical uORFs in a species was generally under purifying selection, we still found a fraction of new canonical uORFs might be favored by positive selection even in primates that typically have a small $N_e$. These observations are consistent with the evolution model of uORFs we previously proposed[4,32]. Under that model, the majority of newly formed uORFs are deleterious and quickly removed from the population, and a relatively smaller fraction of the new uORFs are beneficial and rapidly fixed in populations under positive selection. After fixation, the functional uORFs, particularly the start codons, are maintained by natural selection during evolution. Hence, although in a species the occurrence of uORFs is influenced by positive or purifying selection, the opposing effects of positive selection and purifying selection acting on new uORFs result in a pattern that uORFs are overall depleted in 5′ UTRs. As shown in our population genetic modeling, the efficacies of both positive and purifying selection on uORF fixation in a species are influenced by the effective population size. Moreover, we also found that the gene expression level affects the efficacy of natural selection acting on uORF occurrences. Thus, our results have systematically demonstrated how positive and purifying selection, coupled with differences in gene expression level and $N_e$, influence the genome-wide distribution and contents of uORFs in eukaryotes. Together, our analyses reveal the general principles underlying the distribution and sequence evolution of uORFs in eukaryotes. As uORFs often control posttranscriptional gene

expression in combination with other regulators such as microRNAs[90], further studies are required to elucidate how uORFs coevolve with other regulatory elements.

We found that start codons of canonical uORFs, particularly the translated ones, tend to be maintained by functional constraints during evolution. These results might also be pertinent to the translational buffering mechanism, which indicates that protein expression levels are more conserved between species than mRNAs[116–120]. Nevertheless, our analyses suggest the coding regions of uORFs are overall under neutral evolution. It is not uncommon that some uORF-encoded peptides are conserved across species; however, the conservation of such a peptide does not necessarily mean that peptide might be functional since the coding region of a uORF can be constrained to optimizing translation elongation of that uORF[54,121,122]. Overall, our results suggest that the major function of uORFs is to fine-tune CDS translation rather than to encode conserved peptides. Nevertheless, we do not deny that some uORFs can encode functional peptides, as clearly demonstrated by the previous studies[15,123,124]. Of note, both our PhyloCSF analyses and MS data analyses suggest that a small fraction (<1%) of uORFs might produce peptides.

We found the start codons of the noncanonical uORFs McGillivray et al.[17] identified in humans are overall slightly (~1.2 times) more conserved than the other random triplets across vertebrates. Moreover, our re-analyses of the previously published gene expression data revealed that the noncanonical uORFs tend to have weaker repressive effects on CDS translation than the canonical uORFs, and this pattern was further confirmed by our luciferase reporter assays. Of note, these results do not necessarily suggest that noncanonical uORFs are functionally unimportant, as it has been well established that many noncanonical uORFs might have diverse functions in various biological processes[4,20], such as stress responses[125,126] or tumor initiation[127]. Overall, our current understanding of the prevalence and function of the noncanonical uORFs are still very limited. Further studies are required to reliably identify the noncanonical uORFs and elucidate their regulatory functions and evolutionary principles.

Protists have a very high phylogenetic diversity[128], and many protists use alternative nuclear genetic codes involving stop-codon reassignments[68,69] and obligatory frameshifting at internal stop codons[74]. In protists with no dedicated stop codons[71], such as *Condylostoma magnum*[70,71], *Parduczia* sp.[71], *Blastocrithidia*[72], and *Amoebophrya* sp. ex *Karlodinium veneficum*[73], translation from any possible uAUG is supposed to terminate near the end of a transcript and overlaps with the main CDS, which results in a different protein. Thus, the occurrence of uORFs in protists with alternative genetic decoding schemes might differ considerably from that of most other eukaryotes. In this study, we only focused on 20 protists that use the standard genetic code. Although the O/E ratio of uAUGs was significantly <1 in all the fungi, multicellular plants and animals we examined, such a pattern was observed in only 15 of the 20 protists. The O/E ratio of uAUGs was close to or higher than 1 in the remaining five protists, including *Cystoisospora suis* (1.161, 95% CI 1.154–1.169), *Toxoplasma gondii* (0.998, 95% CI 0.989–1.1.007), *Nannochloropsis gaditana* (0.997, 95% CI 0.986–1.007), and two malaria vectors *Plasmodium yoelii* (1.016, 95% CI 1.008–1.025), and *Plasmodium vivax* (0.989, 95% CI 0.975–1.004). However, these five protists tended to have significantly longer 5′ UTRs than the other 15 protists (Supplementary Fig. S18), suggesting this observation might be an artifact caused by inaccurate 5′ UTR annotations in these five species. Indeed, the O/E ratio of uAUGs in the 5′ UTR regions that are proximal to CDS (within 100 or 150 nt) were significantly lower than 1 in all the five protists (Supplementary Table S5), suggesting that uAUG occurrence in 5′ UTR regions

proximal to CDSs is still under purifying selection in these protists.

The Kozak sequence context around the uoAUG plays a crucial role in controlling the translation of a uORF[16,32,106,107], which subsequently regulates translation of the downstream CDS. There has been a growing interest in engineering uORFs for precise translation control of the main protein products[129–131]. Our results revealed the Kozak sequence context evolved across eukaryotic clades, which suggests that the species-specific Kozak sequence contextual features should be considered in designing uORFs for a specific desired trait.

## Methods

**Identification of putative canonical uORFs.** We downloaded the gene models and cDNA sequences of all eukaryotes that are annotated in the Ensembl Genome Browser (release 96)[132], Ensembl Metazoa (release 43), Ensembl Plants (release 43), Ensembl Protists (release 46), and Ensembl Fungi (release 46). Transcript ends of yeast mRNAs were obtained from a previous study[133]. Putative uORFs and NTEs that start with AUG codons and end with stop codons (UAA/UAG/UGA) were identified from the annotated 5′ UTRs of protein-coding genes. uORFs and NTEs with start codons located in CDSs of other transcripts were excluded from the analysis. Only the species for which 5′ UTR annotation information was available for more than 25% of the protein-coding genes were considered in the analyses. Among all the 479 species meet this criteria, *Ichthyophthirius multifiliis* was excluded since UAA and UAG are reassigned to encode glutamine in this species[134], which would interfere with the uORF and NTE prediction.

**Calculation of the O/E ratio.** A permutation analysis was performed to determine the ratio of the observed to the expected number of uAUGs (O/E ratio) for each species. For genes that exhibited more than one transcript, only the longest transcript was used in the analysis. Unusually long 5′ UTRs in each species (longer than the mean + 3 s.d. of the lengths of the 5′ UTRs in that species) were excluded because these are likely annotation artifacts. We denoted the number of AUG triplets in the 5′ UTRs of a species as $n_{obs}$. We subsequently shuffled the 5′ UTRs with 1000 replicates while maintaining the same dinucleotide frequency using uShuffle[135]. We calculated the median and 2.5% and 97.5% quantiles of the number of AUGs in the shuffled 5′ UTRs and denoted these numbers as $n_{exp}$, $n_{0.025}$, and $n_{0.975}$, respectively. We then calculated the O/E ratios for the species as $n_{obs}/n_{exp}$ with a 95% confidence interval of $[n_{obs}/n_{0.975}, n_{obs}/n_{0.025}]$. The O/E ratio of other triplets in 5′ UTRs or 3′ UTRs was calculated using the same procedure.

**Estimation of the genome-wide ω of protein-coding genes.** To estimate the genome-wide average ω of protein-coding genes between two closely-related species in a clade, we performed a reciprocal best BLAST[136] of protein sequences between the two species ($E < 10^{-10}$). We identified orthologs of protein-coding genes at the genomic scale for 37 pairs of closely related species, which spanned 56 species. For each pair of orthologs between two species, we aligned their protein sequences with MUSCLE (3.8.31)[137] using the default parameters and generated codon alignments with tranalign from the EMBOSS package[138]. We then calculated ω using yn00 from PAML[139] with the codon alignments as input. The median ω of all pairs of orthologs between two species was used as the genome-wide ω of protein-coding genes. For species that were compared with multiple other species, the median ω values obtained from different comparisons were averaged.

**Phylogenetic independent contrasts.** For the 56 metazoan species for which ω values were estimated, we obtained the phylogenetic tree from the Open Tree of Life[140]. We used BUSCO[141] to identify single-copy protein orthologs that were conserved in all 56 species, concatenated the protein sequences of the single-copy orthologs in each species, and performed multiple alignments using MUSCLE with the default parameters. Poorly aligned regions in the resulting alignment were removed using trimAl[142] with the "-automated1" method. The branch length of the tree was calculated using codeml from PAML with the JTT substitution model ("seqtype = 2, runmode = 0, model = 2, aaRateFile = jones.dat"). Phylogenetic independent contrasts were performed using the "pic" function in the ape package[143]. The O/E ratio, ω, and median 5′ UTR length of each species were log-transformed before the contrasts.

**McDonald–Kreitman test of newly fixed uORFs in humans and primates.** To identify fixed differences in AUG triplets in 5′ UTRs and introns, we downloaded whole-genome pairwise alignments between humans (hg19 freeze) and other primates (*Pan troglodytes*, *Gorilla gorilla*, *Pongo abelii*, and *Macaca mulatta*) from the UCSC Genome Browser[144]. AUG triplets that were newly fixed in the human or hominid lineages were inferred using the parsimonious method with *M. mulatta* as the outgroup. We obtained all human SNPs and their ancestral allele information from the phase 3 data of the 1000 Genomes Project[98]. Both fixed and polymorphic AUG differences located in repetitive regions were excluded from the downstream

analysis. Newly fixed or polymorphic AUGs in 5′ UTRs that form NTEs were removed. AsymptoticMK[92,93] tests were performed to detect the signal of positive selection. The data for asymptoticMK tests in flies was obtained from our previous study[32]. To determine the effect of gene expression on positive selection, fixed and segregating mutations were divided into two halves based on the median expression level of genes with fixed new AUGs in 5′ UTR. The average protein abundances across different tissues[145] were used from humans, and the average Reads per Kilobase per Million mapped reads (RPKM) values in Ribo-Seq of 12 different developmental stages or tissues were utilized for flies[32].

**The fixation probability of new uORFs.** For a new autosomal mutation with a selective coefficient s in a diploid population of size $N_e$, the fixation probability of the mutation relative to a neutral mutation was calculated as

$$f(s) = 2N_e \int_0^{\frac{1}{2N_e}} G(x)dx \Big/ \int_0^1 G(x)dx, \text{ where } G(x) = \exp[-4N_e shx - 2N_e s(1 - 2h)x^2]$$

and h is the dominance coefficient[146]. For mutations that introduce new uORFs into the population, the fractions of neutral, deleterious, and beneficial mutations are denoted as $p_1$, $p_2$, and $p_3$, respectively. Based on the assumption that the selective coefficients for deleterious and beneficial mutations have the same absolute value, we can obtain the overall relative fixation probability of mutations as $p_1 + p_2 f(-s) + p_3 f(s)$. In the simulation, we used a fixed $h = 0.5$, and $p_1$, $p_2$, and $p_3$ were set to 0.2, 0.75, and 0.05, respectively.

**Processing of ribosome profiling data.** We obtained pre-calculated ribosome profiling coverage data from humans, mice, rats, zebrafish, and *A. thaliana* from the GWIPS-viz database[85]. Fly ribosome profiling data generated by our group[32] and other researchers[147] that covered all the major developmental stages were also used in this study. Each RPF was assigned to a P-site with plastid[148]. For a uORF in a species, the number of RPFs whose P-sites were within the uORF was calculated with BigWigAverageOverBed[149]. For uORFs that overlapped with CDSs, the overlapping regions were excluded. A uORF was considered as translated if it was covered by the P-site of at least one RPF read across different ribosome profiling datasets in a species.

The genome-wide coverage of initiating ribosome profiling and matched mRNA-Seq data from human and mouse cell lines were downloaded from the GWIPS-viz database. The RNA-Seq data from S2 cells and corresponding Ribo-Seq data after harringtonine treatment were obtained from our previous study[32]. As previously conducted[40], we first counted the number of initiating RPFs whose P-sites were within the 1-nt flanking region (i.e., −1 to +4) of each uORF or CDS start codon and then normalized the initiating RPF count with the mean coverage of the RNA-Seq data in the same region. We only used start codons with at least 2 initiating RPFs and at least 4 mRNA reads for the downstream analysis of human and mouse cell lines, and those with at least 5 initiating RPF reads and at least 10 mRNA reads were used for the analysis of S2 cells.

**Gene ontology analysis.** Gene ontology (GO) annotations for human, mouse, rat, zebrafish, fly, *A. thaliana*, and yeast were downloaded from the Gene Ontology Resource (2019-06-09 release). Because not all genes under a GO term were provided in the GO annotation files, we parsed the gene annotation files to obtain the complete list of genes under each term using topGO[150]. For each species, all the GO terms belonging to Molecular Function, Biological Process, and Cellular Component were combined in the enrichment analysis. The GO terms enriched in uORF-containing genes or uORF-free genes were determined using Fisher's exact tests. Multiple testing correction was performed with the Benjamini–Hochberg method[151], and significant terms were determined at a false discovery rate of 0.1 for each species. Nonredundant representative terms that were significantly enriched in at least five species were chosen for visualization.

**Branch length score calculation.** We downloaded the 100-way vertebrate genome alignments based on human (hg19), 27-way insect alignments based on *D. melanogaster* (dm6), the 7-way alignments of yeast species based on *S. cerevisiae* (sacCer3), and the corresponding phylogenetic trees from UCSC Genome Browser and used the Galaxy platform[152] to parse the multiple sequence alignments of 5′ UTRs in vertebrates or insects. For the start codon of each human uORF (uoAUG), we calculated the sum of the branch lengths of the subtree composed of the species in which the uAUG was present in the orthologous sites ($B_0$) and then calculated the BLS value by dividing $B_0$ by the total branch lengths for the phylogenetic tree of the 100 species. Similarly, the BLS was calculated for the start codon of each uORF in *D. melanogaster* across 27 insect species.

For each predicted uORF peptide in humans, we searched its peptide against the orthologous sequences of other species in the 5′ UTR alignments using Exonerate (V2.2)[153]. uoAUGs located in repeat regions (downloaded from UCSC Genome Browser) were excluded. For oORFs, only the portion that was not overlapping with CDSs were considered in the analysis. To avoid spurious matching, we only considered human uORF peptides containing at least m amino acids (m was set at 10, 15, and 20 in the analysis). We identified uORFs with conserved peptides in other species using the following criteria: (1) the first codon of the matched sequence was AUG; (2) no stop codons or frameshifts were present in the first 80%

of the matched sequence; and (3) between humans and the studied species, the identity of the uORF peptide should be greater than the 2.5% quantile of the genome-wide identity of the main protein sequences. For each uORF, we also calculated the BLSs for peptide sequences based on the presence of conserved peptides in other vertebrates as described above. A similar analysis was performed for the fly and yeast uORFs.

Based on these alignments of uORF peptides, we generated alignments of uORF coding regions between humans and macaques and between *D. melanogaster* and *D. simulans*. Due to the short length of uORFs, we ranked the uORFs based on their Kozak score and divided them into 1000 bins with equal numbers of uORFs. For the uORFs in each bin, we concatenated their alignments and calculated $\omega$ values using yn00 as described above.

**pN/pS analysis**. To study the population variation within uORFs, we merged the genomic intervals of human uORFs and excluded the regions overlapping with CDSs and repeats. We then extracted the SNPs overlapping with uORF regions from the phase 3 data of the 1000 Genomes Project. SNPs in the CDS-overlapping portion of oORFs were excluded. We annotated the effect of SNPs on human uORFs (nonsynonymous or synonymous) using custom scripts and excluded ambiguous SNPs that were annotated as both nonsynonymous and synonymous in different uORFs. For comparison, we also extracted the SNPs in CDS regions and determined their effect on CDSs using SnpEff[154]. The same analysis was performed for uORFs of *D. melanogaster* using the freeze 2 data of the *Drosophila* Genetic Reference Panel[99].

**PhyloCSF score calculation**. The alignments of human uORFs with at least 10 codons were extracted from the 100-way vertebrate genome alignment based on humans as described above. PhyloCSF for each uORF was calculated with Phy-loCSF software[100] using the parameter set "100vertebrates". As a negative control, we annotated all the possible ORFs in 3′ UTRs (dORFs) with at least ten codons using getorf from EMBOSS suit[138]. dORFs overlapping with any CDS or uORF were excluded. We randomly selected 20,000 unique dORFs from the remaining dORFs and calculated PhyloCSF scores with the same procedure as for uORFs. The same analysis was performed for uORFs and dORFs in flies, except that the parameter set "23flies" was used when calculating PhyloCSF scores.

**MS data analysis**. MS datasets for multiple tissues, developmental stages, and cell lines of *D. melanogaster* were obtained from ProteomeCentral[155]. Information on these datasets is listed in Supplementary Data 4. In peptide search, we used a custom database composed of the annotated proteome of *D. melanogaster* and all the peptides encoded by regions between two consecutive in-frame stop codons in cDNA sequences with at least 7 amino acids. To recover as many uORF-encoded peptides as possible, each sample was searched with three different search engines (MaxQuant v1.6.5[156], OpenMS v2.3.0[157], and pFind3[158]) at a 1% false discovery rate. Enzyme specificity was set to trypsin, and at most two missing cleavages were allowed. Cysteine carbamidomethylation was included as the fixed modification and methine oxidation as the variable modification. Both the precursor and fragment tolerance were set to 20 ppm for higher-energy collisional dissociation datasets. Fragment tolerance was set to 0.5 Da for collision-induced dissociation (CID) datasets. Peptides with <7 amino acids were excluded during searching. Peptides that match the built-in contaminants in MaxQuant, yeast proteins and the annotated fly proteome were removed with PeptideMatchCMD v1.0[159] allowing mismatches of leucine and isoleucine. The remaining peptides were mapped to peptides encoded by all the putative canonical uORFs. uORFs with uniquely mapped peptides were kept as MS-supported uORFs.

**Calculation of the Kozak score**. For each species, we retrieved the six nucleotides upstream of the CDS start codons and one nucleotide downstream of these codons and built a position probability matrix (PWM) as the Kozak sequence context. We then determined the Kozak score for the start codon of a uORF or CDS, as well as for each AUG in 3′ UTRs by calculating the log-odds ratio of their flanking sequences using the above-derived PWM[32]. The Euclidian distance between two PWMs of uORF or CDS Kozak sequences was calculated using TFBSTools[160].

**Effect of uORF variation on CDS translation in human populations**. The RNA-Seq and ribosome profiling data from lymphoblastoid cell lines (LCLs) were obtained in a previous study[115]. High-quality genotyping data from 60 LCLs were obtained from the 1000 Genomes project[98]. After pre-processing, the RNA-Seq reads and RPFs were mapped to the human reference genome with STAR[161]. Reads mapped to the CDS region of each protein-coding gene were tabulated with htseq-count[162]. CDS read counts were normalized across different cell lines with DESeq2[163] separately for RNA-Seq and RPFs. The translation efficiency of a gene in a sample was calculated as the ratio of the normalized RPF read count over the normalized RNA-Seq read count. To control for false positives, only SNPs that disrupt the canonical and noncanonical uORFs annotated by McGillivray et al.[17] were analyzed. SNPs with a minor allele frequency of <5% among the 60 LCLs were excluded. A SNP is classified as a canonical uORF variant if the wild-type start codon or mutant start codon is AUG and is classified as noncanonical otherwise. For each uORF variant, a linear regression was performed between the CDS

translational efficiency and the number of non-uORF alleles (0, 1, or 2) across different LCLs.

**Experimental verification of uORF variants**. The effects of uORF variants were assayed with dual-luciferase reporter assays (psiCHECK-2 vector, Promega). HEK293FT cells were purchased from the Cell Bank of the Chinese Academy of Sciences. RNA was extracted using the TRIzol reagent (15596018, Thermo Fisher Scientific), and cDNAs were synthesized using the PrimeScript First-strand cDNA Synthesis Kit (6110B, TaKaRa). For each uORF variant, the wild-type (WT) 5′ UTR or mutated 5′ UTR were cloned from cDNAs by PCR. All the primers used for cloning 5′ UTR fragments were listed in Supplementary Data 7. The reporter plasmid was linearized using Nhe1 (R3131S, NEB). The product and the 5′ UTR sequence were assembled using the NEBuilder HiFi DNA Assembly Cloning Kit (E5520S, NEB). Plasmid libraries were extracted using a QIAGEN Miniprep kit (27106, QIAGEN) according to the manufacturer's instructions. The constructed vectors were transfected into HEK293FT cells using Lipofectamine 3000 Transfection Reagent (L3000015, Thermo Fisher Scientific). The cells were cultivated in Dulbecco's modified Eagle's medium (DMEM) with 10% FBS for 32 h. Then the psiCHECK-2 dual-luciferase reporter assay system (Promega) was used to detect levels of the Renilla luciferase with WT or mutant 5′ UTR and normalized to the firefly luciferase as an internal control. At least four biological repetitions were performed for each WT or mutated 5′ UTR plasmid.

**Reporting summary**. Further information on research design is available in the Nature Research Reporting Summary linked to this article.

## Data availability

The putative uORFs and NTEs annotated in this study are available from figshare[67] (https://doi.org/10.6084/m9.figshare.9980441.v4). The following public data were analyzed in this study: (1) gene annotations, cDNA sequences and genome sequences from Ensembl Genome Browser (https://www.ensembl.org and http://ensemblgenomes.org); (2) the transcript ends of yeast mRNAs from Gene Expression Omnibus (GEO) under the accession number GSE49026; (3) functional annotation of gene categories from The Gene Ontology Resource (http://geneontology.org); (4) gene expression data in model organisms from previous studies as listed in Supplementary Data 2; (5) Ribo-Seq data from GWIPs-viz database (https://gwips.ucc.ie) and our previous study[32]; (6) the effective population size reported in previous studies as listed in Supplementary Table 2; (7) single nucleotide polymorphisms from the 1000 Genomes Project (https://www.internationalgenome.org/data) and DGRP2 (http://dgrp2.gnets.ncsu.edu); (8) multiple genome alignments from UCSC Genome Browser (https://genome.ucsc.edu); (9) the annotation of potential functional uORFs in humans from McGillivray et al.[17]; (10) mass spectrometry datasets from ProteomeCentral (http://proteomecentral.proteomexchange.org) as listed in Supplementary Data 4; 11) RNA-Seq and Ribo-Seq data of human lymphoblastoid cell lines from GEO under the accession number GSE61742 and the Gilad/Pritchard group (http://eqtl.uchicago.edu/RNA_Seq_data). Source Data have been deposited in figshare (https://doi.org/10.6084/m9.figshare.12612068.v2) and are provided with this paper.

## Code availability

The data investigated in this study were analyzed using R statistical software (v3.6). The custom scripts used in this study are available from figshare (https://doi.org/10.6084/m9.figshare.12612068.v2).

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

## Acknowledgements

This work was supported by grants from the National Natural Science Foundation of China (No. 91731301) and the Ministry of Science and Technology of the People's Republic of China (2016YFA0500800) awarded to J.L. and the China Postdoctoral Science Foundation (2019M650003) to Y.W. H.Z. and Y.W. are supported by grants from the National Postdoctoral Innovative Talents Supporting Program. Some of the analyses were performed on the High-Performance Computing Platform of the Center for Life Sciences. The authors thank the National Center for Protein Sciences at Peking University in Beijing, China, for the assistance with the analysis of mass spectrometry data.

## Author contributions

J.L. supervised the entire project and conceived and designed the research. H.Z., Y.W., X.W., X.T., and C.W. contributed to the data analyses. X.W. and X.T. performed the experiments. J.L., H.Z., and Y.W. wrote the manuscript.

## Competing interests

The authors declare no competing interests.
