## [Peer Review File · Nature Communications]

Reviewers' Comments:

Reviewer #1:

Remarks to the Author:

In this manuscript, Zhang et al describe a large scale evolutionary analysis of uORFs across multiple eukaryotic species. Even though the conclusions made based on this analysis are in a general agreement with what we already know about uORFs, the study is unprecedented in its scale and is, therefore, of general interest. Given the intense attention that the topic of small ORFs translation received recently, the manuscript is also timely.

Nonetheless, I found the manuscript to be insufficiently clear in several places. Specifically, I have the following comments.

General:

1. The manuscript gives the impression that it investigates uORFs across all eukaryotes. This, however, is not true. While the study is unprecedented in its scale it is limited to multicellular eukaryotes such as plants and animals. Large phylogenetic clusters, such as fungi and protists are not represented in this study. I strongly suspect that the evolution, distribution and function of uORFs may significantly differ in the organisms from these phyla. Just to give an example, consider recently discovered genetic codes in some ciliates where termination of translation takes place only in close proximity to mRNA 3' ends, e.g. in *Condylostoma magnum* all stop codons code for amino acids in internal positions of mRNA and in *Euplotes* stop codons cause +1 or +2 frameshifting unless in close proximity to the 3' end. In these organisms, once ribosomes initiate translation they are expected to continue the translation of the entire mRNA. Thus, short uORFs are impossible. Given the phylogenetic diversity of many protists and our very limited knowledge of their molecular biology, we may expect many surprising findings regarding the organization of their genetic information and considerable differences from what has been revealed in this manuscript. Thus, at a minimum, the authors should clearly define the phylogenetic boundaries of their study, e.g. "uORFs in plants and animals" instead of "uORFs in eukaryotes", but perhaps it would also be good if the authors discuss the potential limitations of extrapolating their findings on the entire eukaryotic kingdom.

2. One of the authors' conclusions is that most uORFs are regulatory rather than coding for functional peptides, while correctly acknowledging that some uORFs do code for functional peptides. While such a statement is most likely true, it is also vague and hence not very informative. First, "most" stands for "more than a half" which could be 51% or 99%. I wonder if the authors could try to give a more quantitative estimate. In doing so, I also suggest that the authors should take care in defining what they consider functional or perhaps even avoiding the use of the term 'functional', so not to get into a type of controversy such as the one that took place when ENCODE claimed that 80% of the human genome is functional. It seems to me that by function here the authors mean evidence of evolutionary selection. Not all functions are under evolutionary selection, consider human olfactory receptors, many of which, although clearly functional, do accumulate deleterious mutations and evolve almost neutrally. At the same time not all uORFs that exhibit $\omega \ll 1$ necessarily encode functional peptides, because some uORFs are known to alter ribosome movement by making ribosomes stall via specific interactions inside the peptide channel. Such stalling peptides may not function on their own outside of the ribosome even though they would be expected to evolve as protein-coding.

Specific

1. It is not clear how exactly the groups of genes were divided into the categories for the analyses shown in Fig. 1Sb. A more detailed explicit description is necessary.

2. The authors extensively used the data from ref. 27 (McGillivray et al) and attempted to make certain conclusions regarding the evolution of uORFs reported in that work, for example, they found the evidence that these uORFs are more conserved. This is inappropriate. McGillivray et al used conservation as one of the features used for their machine learning algorithm: "Features were chosen to cover a broad range of categories of data, including features associated with uORF position and length, conservation, functional metrics like RNA expression, and sequence-based signatures that may relate to translation." It makes no sense to show increased conservation of uORFs that were predicted based on their conservation.

3. Comparing the strength of Kozak context in uORFs and CDS ATGs. If we take two groups of sequences and compare, they are likely to differ in some respect. If we define one as optimal, the other would become suboptimal. Therefore, the purpose of this analysis is unclear to me. Perhaps it would be more meaningful to compare three groups of contexts rather than two, by adding ATGs that are not used for initiation, e.g. internal ATGs from CDS or ATGs from 3' UTRs or intergenic regions. We expect that the context of such ATGs should not evolve to optimize translation initiation and would provide an estimate for a background context and a variation in contexts. Then we would expect that uORFs context should be optimized for translation initiation, but not as strong as CDS ATG. By having three points the authors could estimate whether uORF ATG context is closer to neutral or that of CDS.

4. "Unsurprisingly, for both uORFs and CDSs, the distance between two species from a clade tended to be significantly shorter than that between one species in that clade and another species outside of that clade (Fig. 6c). These results suggest that the Kozak contextual characteristics tend to be similar between closely related species for both uORFs and CDSs."

This is indeed so unsurprising that it is unclear why was it even done. I believe that any other sequence, e.g. a context of stop codons would exhibit the same behaviour.

5. The authors made an observation that uORFs occurrence anticorrelates with expression levels. This makes sense, but there could be at least two reasons for that. One is that the regulation usually works by suppression, hence the mRNAs whose translation is regulated by uORFs are likely to be lowly expressed. The other is that the negative selection acting on uAUGs is expected to be weaker for lowly expressed mRNAs. These two scenarios are drastically different, could authors try to estimate contributions of each of these two scenarios?

6. To demonstrate the evidence of positive selection on 162 newly fixed uORFs, authors have used the asymptotic McDonald-Kreitman, where the alpha parameter is the proportion of substitutions that are due to adaptive evolution. But confidence intervals are quite wide and contain zeros (as well as negative values), so there seems to be no strong evidence of positive selection (Fig 2a)

7. It is unclear how the relative fixation probability of newly originated uORFs was calculated. Could the authors provide an explicit description of the procedure?

Reviewer #2:

Remarks to the Author:

In the study of Zhang et al., the authors analyzed more than 10 million "uORFs" in over 200 eukaryotic species. They found that 1) most of "uORFs" are under purifying selection. 2) the coding region of "uORFs" is overall less conserved, suggesting that uORF is under neutral evolution or weak selective pressure. Finally, they also analyzed the evolution of start codon and flanking context of uORFs. While the manuscript is written well, many of main conclusions are not new, which have been reported by previous studies. Although previous studies analyzed uORF evolution usually based on a small subset of closely-related species, simply using more species does not

significant extend our knowledge on the origin of uORF translation and its evolution. My major concerns are as follows.

1) Like canonical translation, uORF translation is energy-consuming. Uncontrolled uORF translation may inhibit translation in main CDS. Therefore, it is not unexpected that the potential of uORF translation in 5' UTR has been eliminated during evolution. Also, similar observations have been reported by previous studies, for example PMC5793785, PMC4890304.

2) uORF translation plays various roles in gene expression regulations. As demonstrated by many previous studies (see reviews PMID: 28698598, PMID: 31003826), uORF may encode functional peptide, or uORF translation may control downstream translation in main CDS. Again, it is not unexpected that the coding region of uORFs may not under negative selection, if they do not encode functional peptides.

3) About uORF definition. In this study, uORF is defined as a 5' UTR region starting with ATG and ending with an in-frame stop codon (TAG, TAA or TGA). The uORF definition is problematic. First, they overlooked uORFs starting with non-canonical start codons such as CTG, TTG or ATT. Previous studies have suggested that non-canonical start codons are more prevalence than canonical start codon (i.e. ATG) in uORFs. Second, I would define these regions as putative uORFs or potential uORFs, because majority of these so-called "uORFs" are not translatable. Only a very small number of these putative uORFs are real uORFs with significant protein translation. Analysis based on these putative uORFs will be strongly affected by huge amount of false positives (or background noises), and can not be used to support the conclusions on uORFs.

For example,

i) they found O/E ratio (based on these putative uORFs) is significant lower than 1, suggesting that "purifying selection is the major force shaping the prevalence of uORFs". This result only suggests that ATG triplets are depleted in 5' UTR. Purifying selection for ATG triplets in 5' UTR does not mean a necessary of selection for uORFs. In fact, at least in yeast, a previous study (PMC5793785) reported an elevated non-canonical start codon in 5' UTR, indicating a possibility to maintain some kinds of uORF translation.

ii) The authors found that the dN/dS ratio for uORF CDS is "roughly equal to 1 between human and macaque". They concluded that this result supports neutral evolution of uORFs. However, because majority of "uORFs" in their datasets are non-translatable (or not real uORF), these negative uORFs may significantly increase the dN/dS ratio, since they encode nothing. Again, in *Drosophila*, the dN/dS ratio for all uORF CDSs is close to 1, but later, they found that "uORFs with higher Kozak scores presented significantly lower dN/dS ratio in *Drosophila*, suggesting a scenario in which the coding regions of uORFs with optimal Kozak sequence context are under stronger purifying selection in *Drosophila*". Since ATG surrounded by Kozak sequences are more likely to be translated, I believe their negative result (i.e. dN/dS is close to 1) is due to too many negative uORFs in their datasets.

iii) the same problem can be found in the analysis of "evolution of contextual characteristics that influence uORF translation".

4) Page 4, line 94. "gene expression level is a major determinant of the uORF distribution across genes in a eukaryotic species" Because the number of putative uORFs positively correlates with 5' UTR length, I wondered whether 5' UTR may confound the correlation of putative uORF number to gene expression.

5) Page 4, line 101, "while maintaining the same dinucleotide frequency". Please explain why dinucleotide frequency is maintained. Does single, or trip-nucleotide frequency significantly affect O/E ratio?

6) O/E ratio in 5' UTR might be ok to estimate the selection for ATG triplets. To strength the results, O/E ratio in 3' UTR should be considered as negative control, since translation in 3' UTR ORFs is less likely than that in 5' UTR. In addition, it would be great if O/E ratios for the other 61 triplets are displayed.

7) Page 5, line 122. "The O/E ratio varied wildly across the 216 species", is this ratio affected by different background ATG frequency (E) across the species?

8) Page 8, they found longer uORFs have fewer conserved peptides. This is a little unexpected to me. Because uORF translation is energy-consuming. If a uORF plays regulator role, a shorter ORF is sufficient to block ribosome scanning to downstream region. The longer ORF does not significant

benefit the regulator role, but indeed consume more energy.

**Response to Reviewers' comments**

**Reviewer #1 (Remarks to the Author):**

In this manuscript, Zhang et al describe a large scale evolutionary analysis of uORFs across multiple eukaryotic
species. Even though the conclusions made based on this analysis are in a general agreement with what we
already know about uORFs, the study is unprecedented in its scale and is, therefore, of general interest. Given
the intense attention that the topic of small ORFs translation received recently, the manuscript is also timely.
Nonetheless, I found the manuscript to be insufficiently clear in several places. Specifically, I have the following
comments.

**Response:** We greatly appreciate the enthusiasm and the positive feedback from this reviewer. The comments
and suggestions are precious and very helpful for us to make this revision. In this revision, we have fully
considered your comments and made the revisions accordingly. Please refer to the point-to-point response for
details.

General:

1. The manuscript gives the impression that it investigates uORFs across all eukaryotes. This, however, is not
true. While the study is unprecedented in its scale it is limited to multicellular eukaryotes such as plants and
animals. Large phylogenetic clusters, such as fungi and protists are not represented in this study. I strongly
suspect that the evolution, distribution and function of uORFs may significantly differ in the organisms from
these phyla. Just to give an example, consider recently discovered genetic codes in some ciliates where
termination of translation takes place only in close proximity to mRNA 3' ends, e.g. in *Condylostoma magnum*
all stop codons code for amino acids in internal positions of mRNA and in *Euplotes* stop codons cause +1 or +2
frameshifting unless in close proximity to the 3' end. In these organisms, once ribosomes initiate translation
they are expected to continue the translation of the entire mRNA. Thus, short uORFs are impossible. Given the
phylogenetic diversity of many protists and our very limited knowledge of their molecular biology, we may
expect many surprising findings regarding the organization of their genetic information and considerable
differences from what has been revealed in this manuscript. Thus, at a minimum, the authors should clearly
define the phylogenetic boundaries of their study, e.g. "uORFs in plants and animals" instead of "uORFs in
eukaryotes", but perhaps it would also be good if the authors discuss the potential limitations of extrapolating
their findings on the entire eukaryotic kingdom.

**Response:** We thank this reviewer for pointing out this issue. In this revision, we added 242 fungi and 23 protists
into our analysis to cover all the large phylogenetic clusters of eukaryotes. The annotated putative canonical
uORFs in fungi and protists are summarized as follows (Page 4, Lines 78-84):

"The number of annotated protein-coding genes in the 242 fungus genomes ranged from 3,623
(*Pneumocystis murina*) to 32,847 (*Fibularhizoctonia sp.*). We identified a total of 3,469,095 uORFs in these
fungal genomes, with the number of uORFs ranging from 1,233 (*Malassezia sympodialis*) to 94,695
(*Verticillium longisporum*) (Supplementary Table 1). Among the 23 protists, the number of annotated protein-
coding genes ranged from 3,398 (*Condylostoma magnum*) to 38,544 (*Emiliana huxleyi*), and the number of

uORFs ranging from 1,903 (*Plasmodium falciparum*) to 99,859 (*Cystoisospora suis*), which resulted in a total
of 434,267 uORFs in these protist genomes (Supplementary Table 1).”

The downstream analyses including the O/E ratio comparisons (Supplementary Figs. 2-4), the influence of gene
expression on uORF occurrences (Supplementary Fig. 5), selective constraints on uORF sequences
(Supplementary Fig. 8 and 10), and the start codon sequence context (Fig. 6; Supplementary Fig. 13 and 16)
was performed for these newly included species as well. Please refer to the updated analyses in TEXT.

Although the patterns of uORF distribution and sequence evolution in fungi and protists are largely
consistent with that in multicellular animals and plants, some differences were indeed observed in several
protists. However, our analysis suggests that the overall occurrence of uORFs are still under purifying selection
in these species (Page 14, Lines 453-466):

“Among the 481 eukaryotes we studied, the O/E ratio of uORFs was significantly less than 1 in all the 216
multi-cellular and the 242 fungal species. Such a pattern was observed in only 17 of the 23 protists, however,
and the O/E ratio of uORFs was close to or higher than 1 in the remaining six protists, including *Condylostoma*
*magnum* (1.041, 95% CI 1.031~1.050), *Cystoisospora suis* (1.161, 95% CI 1.154~1.169), *Toxoplasma gondii*
(0.998, 95% CI 0.989~1.1.007), *Nannochloropsis gaditana* (0.997, 95% CI 0.986~1.007), and two malaria
vectors *Plasmodium yoelii* (1.016, 95% CI 1.008~1.025) and *Plasmodium vivax* (0.989, 95% CI 0.975~1.004).
It is well established that in protists the nuclear genetic code frequently changed, mostly due to stop codon
reassignments¹. In particular, *C. magnum* has no dedicated stop codons², and every uORF is supposed to
terminate near the end of a transcript and overlaps with the main CDS. Interestingly, the O/E ratio of uORFs in
the 5' UTR regions that are proximal to CDS (within 100 nt or 150 nt) were significantly lower than 1 in five of
the six protists except *C. magnum* (Supplementary Table 11). Thus, our results suggest that overall uORF
occurrence in 5' UTRs of protists is still under purifying selection, however, whether and how the genetic code
reassignments affect the distribution and evolution of uORFs in certain protists deserve further study.”

2. One of the authors' conclusions is that most uORFs are regulatory rather than coding for functional peptides,
while correctly acknowledging that some uORFs do code for functional peptides. While such a statement is
most likely true, it is also vague and hence not very informative. First, “most” stands for “more than a half”
which could be 51% or 99%. I wonder if the authors could try to give a more quantitative estimate. In doing so,
I also suggest that the authors should take care in defining what they consider functional or perhaps even
avoiding the use of the term ‘functional’, so not to get into a type of controversy such as the one that took place
when ENCODE claimed that 80% of the human genome is functional. It seems to me that by function here the
authors mean evidence of evolutionary selection. Not all functions are under evolutionary selection, consider
human olfactory receptors, many of which, although clearly functional, do accumulate deleterious mutations
and evolve almost neutrally. At the same time not all uORFs that exhibit $\omega \ll 1$ necessarily encode
functional peptides, because some uORFs are known to alter ribosome movement by making ribosomes stall
via specific interactions inside the peptide channel. Such stalling peptides may not function on their own outside
of the ribosome even though they would be expected to evolve as protein-coding.

**Response:** These comments are enlightening and much appreciated. To address these concerns, we performed
two additional analyses in this revised version. First, we performed PhyloCSF analysis of the coding regions
of uORFs. The PhyloCSF algorithm predicts whether a genomic region potentially represents a conserved
protein-coding region or not based on multiple sequence alignments³, and a positive PhyloCSF score means that
region is more likely to encode a peptide. As negative controls, we also calculated the PhyloCSF scores for
20,000 randomly selected ORFs in 3' UTRs (downstream ORFs, dORFs), as these dORFs have little chance of
translation. By comparing the PhyloCSF scores of the uORFs that showed evidence of translation with the
ribosome profiling data versus the random dORFs, we estimated that in humans, 0.44% (161 out of 36,655) of
the translated uORFs showed evidence of encoding conserved peptides, and that in *Drosophila*, 0.80% (102 of
12,754) of the translated uORFs might encode conserved peptides. Overall, these analyses suggest that less than
1% canonical uORFs might encode conserved peptides.

Next, we examined public mass spectrometry (MS) datasets for evidence of uORF-encoded peptides in
*Drosophila*. We analyzed the mass spectrometry (MS) data from 38 samples of different developmental stages
or tissues of *D. melanogaster* from previous studies⁴⁻⁸ (see Supplementary Table 7 for details). Among the
24,462 uORFs that met our parameter settings, 84 (0.34%) had peptides detected in at least one sample (see
Supplementary Table 8 for details). In combination with our finding that most uORFs do not encode conserved
peptides, these results suggest that only a very small fraction (< 1%) of the uORFs might encode peptides that
are maintained by natural selection during evolution.

The new results are described as follows (Page 10, Lines 318-338):

“To estimate the proportion of uORFs that might encode conserved peptides, for each uORF, we also
calculated PhyloCSF score, which predicts whether a genomic region potentially represents a conserved protein-
coding region or not based on multiple sequence alignments³ (a positive PhyloCSF score means that region is
more likely to encode a peptide). As a negative control, we also calculated the PhyloCSF scores for 20,000
randomly selected ORFs in 3' UTRs (downstream ORFs, dORFs), as these dORFs have little chance of
translation. Among the 36,655 uORFs that are ≥ 10 codons and evidenced of translation in humans, only 361
(0.985%) had positive PhyloCSF scores (Supplementary Fig. 12a). In contrast, the PhyloCSF score was positive
for 0.545% (109 out of 20,000) dORFs. Thus, after controlling for the background noises, only 0.44% (161) of
the translated uORFs showed evidence of encoding conserved peptides. In *Drosophila*, 1.19% (152 of 12,745)
translated uORFs and 0.39% (78 out of 20000 dORFs) had positive PhyloCSF scores, yielding an estimate of
0.80% (102 of 12,754) of the translated uORFs might encode conserved peptides. Overall, these analyses
suggest that less than 1% canonical uORFs might encode conserved peptides.

To test whether our evolutionary analyses of uORFs were supported by experimental evidence, we
analyzed the mass spectrometry (MS) data from 38 samples of different developmental stages or tissues of *D.*
*melanogaster* (Supplementary Table 8)⁴⁻⁸. Among the 24,462 uORFs that met our parameter settings (Methods),
84 (0.34%) had peptides detected in at least one sample (Supplementary Table 9). Interestingly, the BLS
analysis revealed that the MS-supported uORFs present more conserved coding regions than the other uORFs
(Fig. 5e), suggesting these MS-supported uORF peptides might be functionally important. Collectively, our
results support the notion that most uORFs play regulatory roles and their start codons are maintained due to

functional constraints, and only a tiny fraction (< 1%) of the uORFs might encode peptides that are maintained
by natural selection during evolution.”

We also put the following sentences in Discussion (Page 13, Lines 438-442), which are reproduced as follows:

“Overall, our results suggest that the major function of uORFs is to fine-tune CDS translation rather than
to encode conserved peptides. Nevertheless, we do not deny that some uORFs can encode functional peptides,
as clearly demonstrated by the previous studies⁹⁻¹¹. Of note, both our PhyloCSF analyses and MS data analyses
suggest that a small fraction (< 1%) of uORFs might produce peptides.”

Specific

1. It is not clear how exactly the groups of genes were divided into the categories for the analyses shown in Fig.
1Sb. A more detailed explicit description is necessary.

**Response:** We apologize that this information related to this figure (Supplementary Fig. 5b in the revised
manuscript) was not clearly described in our previous version. In this revised version, we presented the details
in the “Gene ontology analysis” subsection of Methods, which is reproduced as follows (Page 17, Lines 552-
560):

“Gene ontology annotations for human, mouse, rat, zebrafish, fly, *A. thaliana*, and yeast were downloaded
from the Gene Ontology Resource (2019-06-09 release). Because not all genes under a GO term were provided
in the GO annotation files, we parsed the gene annotation files to obtain the complete list of genes under each
term using topGO¹². For each species, all the GO terms belonging to Molecular Function (MF), Biological
Process (BP), and Cellular Component (CC) were combined in the enrichment analysis. The GO terms that
were enriched in uORF-containing genes or uORF-free genes were determined using Fisher's exact tests.
Multiple testing correction was performed with the Benjamini-Hochberg method¹³, and significant terms were
determined at a false discovery rate (FDR) of 0.1 for each species. Non-redundant representative terms that
were significantly enriched in at least five species were chosen for visualization.”

We also updated the figure legend (Supplementary Fig. 5b in the revised manuscript), which is reproduced
as follows:

“Gene categories enriched in uORF-free genes (left) and uORF-containing genes (right). In each species,
genes belonging to each category were extracted from the annotations provided by Gene Ontology Consortium
(Methods). Whether a gene category is enriched in the gene set is assessed with Fisher's exact test. Multiple
testing correction was performed with the Benjamini-Hochberg method¹³. The odds ratios (log2) and adjusted
*P* values are indicated by the color and size of the points, respectively. For uORF-containing genes, the same
analysis was performed for all the genes containing putative uORFs or only genes containing translated uORFs.
Non-redundant representative terms that are enriched in at least five of the seven model organisms were
displayed in the plot. See Supplementary Table 3 for the complete list of terms enriched in each species. Some
terms are insignificant in yeast, primarily because yeast is a unicellular organism with only 6,600 protein-coding
genes and 955 of those genes contain uORFs, which makes the statistical power of enrichment analyses
relatively low in yeast.”

2. The authors extensively used the data from ref. 27 (McGillivray et al) and attempted to make certain
conclusions regarding the evolution of uORFs reported in that work, for example, they found the evidence that
these uORFs are more conserved. This is inappropriate. McGillivray et al used conservation as one of the
features used for their machine learning algorithm: “Features were chosen to cover a broad range of categories
of data, including features associated with uORF position and length, conservation, functional metrics like RNA
expression, and sequence-based signatures that may relate to translation.” It makes no sense to show increased
conservation of uORFs that were predicted based on their conservation.

**Response:** Thank you for raising this concern. In our previous version, we mainly focused on the evolution of
canonical uORFs. We did not emphasize the evolutionary patterns of the noncanonical uORFs because 1) most
species surveyed in this study currently have no ribosome profiling data and, 2) it is very challenging to predict
the noncanonical uORFs *in silico* reliably. Hence, in our previous submission, we did not intend to show that
the noncanonical uORFs are more conserved. We touched the predicted noncanonical uORFs from McGillivray
*et al.*¹⁴ mainly in two places. First, when comparing the conservation of uAUGs relative to the background
triplets in the 5' UTRs, we excluded the start codons of 173,290 noncanonical uORFs (beginning with non-
AUG triplets) identified by McGillivray *et al.* from the backgrounds. Second, we found that the noncanonical
uORFs predicted in the previous study were indeed slightly more conserved than random triplets, but these are
significantly less conserved than the canonical uORFs.

We agree with the reviewer that it is inappropriate to compare the conservation of noncanonical uORFs
predicted based on conservation to the remaining random triplets. We rephrased the relevant sentences which
read as follows (Page 9, Lines 264-267): “We also calculated the BLS values for the start codons of the 173,290
noncanonical uORFs previously identified in humans by McGillivray *et al.*¹⁴. Since conservation was used as a
feature to identify the noncanonical uORFs in that study, it is not surprising that these noncanonical start codons
were slightly (~1.2 times) more conserved than the other random triplets ($P=2.1\times 10^{-77}$, WRST; Fig. 3d).
However, they were significantly less conserved than the canonical uAUGs ($P=1.0\times 10^{-10}$, WRST).”

Moreover, in the revised manuscript, we have added new analyses regarding the biological functions of
noncanonical uORFs from two aspects. First, we extracted previously published functional and population
genomic data and examined whether variations in uORF start codons influence the translation efficiency of the
main CDSs among different samples (Fig. 7a). Among the potentially functional uORFs in humans predicted
by McGillivray *et al.*¹⁴, 146 canonical and 796 noncanonical uORFs had genetic variants in their start codons
among these samples (only variants with minor allele frequency $\geq 5\%$ were considered in the analysis). We
performed linear regressions to assess the regulatory impact of uORF alteration on the translation of down-
stream CDSs, with a positive slope value in the regression meaning that the presence of a uORF in certain
individuals is associated with a decrease in the translation efficiency of the downstream CDS in those
individuals, and vice versa. A general trend was the slope values were overall positive for the canonical uORFs,
while the slope values for the noncanonical uORFs fluctuated around 0 (Fig. 6b). This comparison suggests that
in human populations, the noncanonical uORFs overall have relatively limited repressive effects on CDS
translation compared to the canonical uORFs.

Next, we experimentally verified the influence of both types of uORFs on CDS translation. We sampled

80 human uORFs and performed luciferase reporter assays in HEK293FT cells (Supplementary Fig. 17). These
tested uORFs, which included 42 canonical and 38 noncanonical ones, were predicted potentially functional by
McGillivray *et al.*¹⁴ and had polymorphic start codons in human populations. For each uORF, we compared the
repressive effect of the annotated uORF allele versus that of the non-uORF allele in suppressing the translation
of the reporter gene. Although occasionally the non-uORF allele had a stronger repressive effect than the uORF
allele, the general trend was that the uORF allele had a stronger effect than the non-uORF allele in suppressing
translation (Fig. 6c-d). Moreover, a significantly higher proportion of the canonical (55%, 23/42) than the
noncanonical (26%, 10/38) uORFs exhibited the pattern that the annotated uORF allele showed a significantly
stronger repressive effect on the CDS translation than the non-uORF allele ($P = 0.013$, Fisher's exact test, Fig.
6c-d). Also, the difference in CDS translation suppression between the uORF and the non-uORF allele is
significantly larger for the canonical than the noncanonical uORFs ($P = 0.006$, WRST). Altogether, these results
reinforced the thesis that the noncanonical uORFs overall have weaker repressive effects on CDS translation
than the canonical uORFs. We fully described these new results in the Section “**Comparing the canonical
versus noncanonical uORFs in repressing CDS translation in human populations**” (Page 12, Lines 379-
412).

3. Comparing the strength of Kozak context in uORFs and CDS ATGs. If we take two groups of sequences and
compare, they are likely to differ in some respect. If we define one as optimal, the other would become
suboptimal. Therefore, the purpose of this analysis is unclear to me. Perhaps it would be more meaningful to
compare three groups of contexts rather than two, by adding ATGs that are not used for initiation, e.g. internal
ATGs from CDS or ATGs from 3' UTRs or intergenic regions. We expect that the context of such ATGs should
not evolve to optimize translation initiation and would provide an estimate for a background context and a
variation in contexts. Then we would expect that uORFs context should be optimized for translation initiation,
but not as strong as CDS ATG. By having three points the authors could estimate whether uORF ATG context
is closer to neutral or that of CDS.

**Response:** Thank you for the constructive suggestions. In this revised version, we followed this reviewer's
suggestion and used the AUGs in 3' UTRs (dAUGs) as negative controls to discriminate whether the context of
uAUGs is optimized or close to the neutral background. We reported the new results as follows (Page 11, Lines
362-268):

“To test whether the sequence contexts of uAUGs are optimized, in each species we also calculated the
Kozak scores of the AUG triplets in 3' UTRs (downstream AUGs, dAUGs) as neutral controls. The Kozak
scores of uAUGs were significantly higher than those of dAUGs in most (88.2%, 120 out of 136) vertebrates,
(68.3%, 28 out of 41) plants and (180%, 180 out of 242) fungi; however, an opposite trend was observed in
invertebrates and no obvious trend was observed in protists (Supplementary Fig. 16b). These results suggest
that the optimization of the Kozak sequence context of uORFs is different across eukaryotic clades.”

We did not include the Kozak scores for the AUG triples inside the coding regions because the -6 to +4
nucleotides around such internal AUGs are under strong selective constraints due to coding functions or codon
usage bias.

4. “Unsurprisingly, for both uORFs and CDSs, the distance between two species from a clade tended to be
significantly shorter than that between one species in that clade and another species outside of that clade (Fig.
6c). These results suggest that the Kozak contextual characteristics tend to be similar between closely related
species for both uORFs and CDSs.”

This is indeed so unsurprising that it is unclear why was it even done. I believe that any other sequence, e.g. a
context of stop codons would exhibit the same behavior.

**Response:** We thank the reviewer for raising this concern. The aim of our analysis is to explore whether the
Kozak contexts around the start codons of uORFs are different across eukaryotic clades. To our knowledge,
such an issue has not been systematically explored yet. We think this question is important as there is growing
interest in engineering uORFs for precise translation control of the main protein products. Our results suggest
that considering species-specific Kozak sequence contextual features might be necessary in designing uORFs
for a specific desired trait in a certain species.

In this revised manuscript, we emphasized this point in Discussion (Page 14, Lines 468-471) with the
following sentences:” There has been a growing interest in engineering uORFs for precise translation control
of the main protein products¹⁵⁻¹⁷. Our results revealed the Kozak sequence context evolved across eukaryotic
clades, which suggests that the species-specific Kozak sequence contextual features should be considered in
designing uORFs for a specific desired trait.”

5. The authors made an observation that uORFs occurrence anticorrelates with expression levels. This makes
sense, but there could be at least two reasons for that. One is that the regulation usually works by suppression,
hence the mRNAs whose translation is regulated by uORFs are likely to be lowly expressed. The other is that
the negative selection acting on uAUGs is expected to be weaker for lowly expressed mRNAs. These two
scenarios are drastically different, could authors try to estimate contributions of each of these two scenarios?

**Response:** These comments are really insightful and enlightening. To address this issue, in this revised version,
we grouped genes of a species into 20 equal-sized bins based on increasing expression levels and calculated the
O/E ratio of uORFs in each bin. In all the five species we examined, the O/E ratio was substantially lower than
1 in each bin (Supplementary Fig. 5c), suggesting that purifying selection was the dominant evolutionary force
acting on the uORF occurrence regardless of gene expression levels. Nevertheless, we observed significant
anticorrelations between the gene expression level and O/E ratio of uORFs in each species, suggesting that
purifying selection acting on uAUGs is relatively weak for lowly expressed genes. On the other hand, genes in
certain functional categories, such as transcriptional factors, which are likely to be lowly expressed, might be
preferentially suppressed by uORFs at the translational level for optimizing protein production. Thus, our results
suggest that although gene expression level overall is an important factor influencing the genome-wide
distributions of uORFs across genes, the anticorrelation between gene expression level and uORF occurrence
is caused by very complex factors.

In this revised version, we have re-written the relevant section, which is reproduced as follows (Page 5,
Lines 139-167):

**“Gene expression level as an important factor influencing the genome-wide distributions of uORFs across**
**genes**

In humans, genes with uORFs exhibited lower expression levels than genes without uORFs¹⁸. Similarly, our
analysis of previously published mRNA and protein abundance data of fly, human, mouse, mustard plant, and
yeast revealed uORFs were infrequently detected in housekeeping genes, and there were significant
anticorrelations between the gene expression level and the number of uORFs (Supplementary Fig. 5a and
Supplementary Table 2). Meanwhile, gene ontology analysis revealed that genes containing putative uORFs
tend to be enriched in the categories of signal transduction, transcription factors, and membrane proteins
(Supplementary Fig. 5b; Supplementary Table 3). These patterns still held when we focused on the uORFs
supported by previously published ribosome profiling data in fly¹⁹ and other species collected in the GWIPs-
viz database²⁰ (Supplementary Table 4). Noteworthy, the anticorrelation between uORF occurrences and gene
expression level well reconciles with the gene ontology analyses as housekeeping genes tend to be highly (or
broadly) expressed²¹.

Since gene expression level affects the efficacy of natural selection²², we further asked whether the efficacy
of purifying selection is reduced in removing deleterious uORFs in lowly expressed genes. We grouped genes
of a species into 20 equal-sized bins based on increasing expression levels and calculated the O/E ratio of uORFs
in each bin. In all the five species we examined, the O/E ratio was lower than 1 in each bin (Supplementary Fig.
5c), suggesting that purifying selection was the dominant evolutionary force acting on the uORF occurrence
regardless of gene expression levels. Interestingly, we observed significant anticorrelations between the gene
expression level and O/E ratio of uORFs in each species, suggesting that purifying selection acting on uAUGs
is relatively weak for lowly expressed mRNAs.

Thus, our results suggest that gene expression level is an important factor influencing uORF distribution
across genes in a eukaryotic species. It is possible that excessive uORFs in highly expressed genes might cause
insufficient protein output, which is harmful to the organisms. We postulate that purifying selection has removed
deleterious uORFs in the highly expressed genes more efficiently than in the lowly expressed genes. On the
other hand, genes in certain functional categories, such as transcriptional factors, which are likely to be lowly
expressed, might be preferentially suppressed by uORFs at the translational level for optimizing protein
production. Further studies are needed to investigate the relative importance of the two mechanisms in shaping
the anticorrelation between gene expression level and uORF occurrence.”

6. To demonstrate the evidence of positive selection on 162 newly fixed uORFs, authors have used the
asymptotic McDonald-Kreitman, where the alpha parameter is the proportion of substitutions that are due to
adaptive evolution. But confidence intervals are quite wide and contain zeros (as well as negative values), so
there seems to be no strong evidence of positive selection (Fig 2a)

**Response:** We greatly appreciate this point. In this revision, we updated the analysis and made two interesting
observations. First, we detected strong and significant positive selection for newly fixed ATGs derived from

CpG to TpG mutations in primates. Second, we found that the signal of positive selection is more pronounced
in the genes with higher expression levels, both in primates and in *Drosophila*. In fact, the new results were
largely inspired by the enlightening comments this reviewer raised regarding the gene expression level in the
previous point. We reported the new results as follows (Page 7, Lines 210-225):

“We detected weak signals of positive selection on the newly fixed uORFs in all three branches, and the
value of α_{asym} , which represents the fraction of newly formed uORFs driven to fixation by positive selection,
was 0.18 (95% CI, -0.15~0.50), 0.15 (95% CI, -0.20~0.48), and 0.14 (95% CI, -0.21~0.48) in the three branches,
respectively (Fig. 2a). Noteworthy, C>T mutations at CpG dinucleotides are highly frequent in mammals²³, and
new AUGs can be generated from CpG to TpG mutations through two approaches²⁴: 1) from ACG to ATG, and
2) from CGTG to CATG (Fig. 2b). Thus, we further examined new uORFs derived from the CpG contexts and
the remaining new uORFs separately. Roughly speaking, ~33% of the new AUGs fixed in each of the three
branches were generated by CpG to TpG mutations. Interestingly, the CpG-derived uORFs were under strong
positive selection (the α_{asym} was 0.57 (95% CI, 0.29~0.85), 0.54 (95% CI, 0.24 ~ 0.83) and 0.53 (95% CI, 0.22~
0.83) in the three branches, respectively), while the α_{asym} for the remaining uORFs was close to 0 (Fig. 2b).
Noteworthy, the α_{asym} values were even higher when we focused on the new uORFs that were derived from the
CpG contexts in the highly expressed genes (Supplementary Table 6). Of note, for the new uORFs fixed in *D.*
*melanogaster* we previously analyzed¹⁹, a higher α_{asym} value was also observed for the highly expressed genes
(Supplementary Table 7). Therefore, although the prevalence of uORFs in a species was generally under
purifying selection, we still found a fraction of uORFs might be favored by positive selection even in primates
that typically have a small N_e .”

7. It is unclear how the relative fixation probability of newly originated uORFs was calculated. Could the authors
provide an explicit description of the procedure?

**Response:** We apologize for the obscure description of this process. In this revised manuscript, we have
provided a detailed description of the procedure used to calculate the relative fixation probability in “Methods”
(Page 14, Lines 464-472):

**“The fixation probability of new uORFs**

For a new autosomal mutation with a selective coefficient s in a diploid population of size N_e , the fixation
probability of the mutation relative to a neutral mutation was calculated as $f(s) =$

$2N_e \int_0^{\frac{1}{2N_e}} G(x)dx / \int_0^1 G(x)dx$, where $G(x) = \exp[-4N_e s h x - 2N_e s(1 - 2h)x^2]$ and h is the dominance

coefficient²⁵. For mutations that introduce new uORFs into the population, the fractions of neutral, deleterious,
and beneficial mutations are denoted as p_1 , p_2 , and p_3 , respectively. Based on the assumption that the selective
coefficients for deleterious and beneficial mutations have the same absolute value, we can obtain the overall
relative fixation probability of mutations as $p_1 + p_2 f(-s) + p_3 f(s)$. In the simulation, we used a fixed $h=0.5$,
and p_1 , p_2 , and p_3 were set to 0.2, 0.75, and 0.05, respectively.”

**Reviewer #2 (Remarks to the Author):**

In the study of Zhang et al., the authors analyzed more than 10 million “uORFs” in over 200 eukaryotic species.
They found that 1) most of “uORFs” are under purifying selection. 2) the coding region of “uORFs” is overall
less conserved, suggesting that uORF is under neutral evolution or weak selective pressure. Finally, they also
analyzed the evolution of start codon and flanking context of uORFs. While the manuscript is written well,
many of main conclusions are not new, which have been reported by previous studies. Although previous studies
analyzed uORF evolution usually based on a small subset of closely-related species, simply using more species
does not significant extend our knowledge on the origin of uORF translation and its evolution. My major
concerns are as follows.

**Response:** We thank this reviewer for the thorough reviews. In this manuscript, we indeed found that 1) most
of “uORFs” are under purifying selection, and 2) the coding region of “uORFs” is overall less conserved, which
has been nicely summarized by this reviewer. **However, we went steps further than merely reporting such**
**observations.** The novel findings relevant to these two points are 1) we have demonstrated how positive and
purifying selection, coupled with differences in gene expression level and N_e , influence the genome-wide
distribution and contents of uORFs in eukaryotes, and 2) the uAUGs, particularly the translated uAUGs, tend
to be maintained by functional constraints during evolution, however, the coding regions of uORFs are overall
under neutral evolution. This comparison suggests that the major function of uORFs is to fine-tune CDS
translation rather than encode conserved peptides. In this new submission, inspired by the comments from this
reviewer, we also performed new analyses to highlight the novel findings of this study. For example, we further
carried out both phyloCSF analyses and mass spectrometry (MS) data analyses to demonstrate that only a small
fraction (< 1%) of uORFs might produce peptides (Page 10, Lines 318-338).

We believe the manuscript is much improved after addressing this reviewer’s concerns. We also highlighted
our changes in the Point-to-point response section.

1) Like canonical translation, uORF translation is energy-consuming. Uncontrolled uORF translation may
inhibit translation in main CDS. Therefore, it is not unexpected that the potential of uORF translation in 5’ UTR
has been eliminated during evolution. Also, similar observations have been reported by previous studies, for
example PMC5793785, PMC4890304.

**Response:** We agree with this reviewer that it is not surprising to find that uORFs are generally depleted in 5’
UTRs by natural selection. However, how the efficiency of purifying selection and positive selection acting on
uORFs prevalence differs across species or genes have not been thoroughly explored prior to this study.
Specifically, our novel discoveries are summarized as follows:

1) With the comparative analysis of uORF occurrences in 481 eukaryotes (242 fungi and 23 protists were newly
included in this revision), we found the trend of uORF depletion varies widely across different species, and the
degree of uORF depletion is mainly determined by the effective population size of a species.

2) Previously, we found that newly fixed uORFs in *D. melanogaster* are under positive selection. Here, we
extend this analysis to primates and found a similar trend, which suggests that although uORFs are overall under

depletion, a fraction of uORFs are favored by positive selection even in species with a small effective population
size such as primates. In particular, we detected strong and significant positive selection for newly fixed ATGs
derived from CpG to TpG mutations in primates. Moreover, we found that the signal of positive selection is
more pronounced in the genes with higher expression levels, both in primates and in *Drosophila*. We present
the updated results in Page 7, Lines 210-225.

3) We investigated the factors that influence the efficiency of uORF depletion within a species. We found that
the gene expression level is one major factor that influences uORF distribution across genes, and that purifying
selection on uORF prevalence is stronger in highly expressed genes. We present the new results in Page 6, Lines
152-167.

4) Based on the position and frame of a uORF relative to the downstream CDS, in this revision, we classified
uORFs into nonoverlapping uORFs, out-of-frame overlapping uORFs (oORFs), and N-terminal extensions. We
found that the O/E ratio of oORFs was overall significantly lower than that of nonoverlapping uORFs
(Supplementary Fig. 4). Interestingly, N-terminal extensions showed the lowest O/E ratio among the three
categories of uORFs in 460 out of 481 species (Supplementary Fig. 4), suggesting that novel N-terminal
extensions might be harmful to normal protein functions and tend to be depleted. We present the new results in
Page 5, Lines 122-128.

Given the above considerations, we believe that our study provides novel insights into the distribution and
evolution of uORFs in eukaryotes beyond the general idea that uORFs tend to be depleted from 5' UTR during
evolution.

2) uORF translation plays various roles in gene expression regulations. As demonstrated by many previous
studies (see reviews PMID: 28698598, PMID: 31003826), uORF may encode functional peptide, or uORF
translation may control downstream translation in main CDS. Again, it is not unexpected that the coding region
of uORFs may not under negative selection, if they do not encode functional peptides.

**Response:** Thank you for pointing this out. We agree with the reviewer that the coding region of an uORF is
likely not under negative selection if it does not encode functional peptides. However, the type of uORF
functions (regulatory versus coding) that dominate has not been thoroughly investigated. In this study, we
performed comparative studies of uORF start codons and coding regions through phylogenetic and population
genetic analyses. Our results suggest that although uORF start codons are more conserved than expected, the
coding regions of uORFs usually evolve neutrally or under weak selective constraints, which leads us to
conclude that most uORFs do not encode conserved peptides.

To estimate the proportion of uORFs that might encode conserved peptides, for each uORF, we also
calculated the PhyloCSF score, which predicts whether a genomic region potentially represents a conserved
protein-coding region or not based on multiple sequence alignments. Our PhyloCSF analysis revealed that only
0.44% of the human uORFs that are evidenced of translation might encode conserved peptides, and similarly,
0.80% (102 of 12,754) of the translated uORFs in *Drosophila* might encode conserved peptides. Overall, these
analyses suggest that less than 1% canonical uORFs might encode conserved peptides.

However, functional uORFs are not necessarily conserved. Therefore, we also searched for uORF-encoded
peptides among multiple mass spectrometry datasets in *D. melanogaster*. We found MS evidence for only 84
(0.23%) uORFs, and the peptides encoded by MS-supported uORFs are more conserved than those of the
remaining uORFs. Taken together, our results support the notion that the dominant function of uORFs is
regulatory rather than encoding peptides. The new analysis is described as follows (Page 10, Lines 318-338):

“To estimate the proportion of uORFs that might encode conserved peptides, for each uORF, we also
calculated PhyloCSF score, which predicts whether a genomic region potentially represents a conserved protein-
coding region or not based on multiple sequence alignments³ (a positive PhyloCSF score means that region is
more likely to encode a peptide). As a negative control, we also calculated the PhyloCSF scores for 20,000
randomly selected ORFs in 3' UTRs (downstream ORFs, dORFs), as these dORFs have little chance of
translation. Among the 36,655 uORFs that are ≥ 10 codons and evidenced of translation in humans, only 361
(0.985%) had positive PhyloCSF scores (Supplementary Fig. 12a). In contrast, the PhyloCSF score was positive
for 0.545% (109 out of 20,000) dORFs. Thus, after controlling for the background noises, only 0.44% (161) of
the translated uORFs showed evidence of encoding conserved peptides. In *Drosophila*, 1.19% (152 of 12,745)
translated uORFs and 0.39% (78 out of 20000 dORFs) had positive PhyloCSF scores, yielding an estimate of
0.80% (102 of 12,754) of the translated uORFs might encode conserved peptides. Overall, these analyses
suggest that less than 1% canonical uORFs might encode conserved peptides.

To test whether our evolutionary analyses of uORFs were supported by experimental evidence, we analyzed
the mass spectrometry (MS) data from 38 samples of different developmental stages or tissues of *D.*
*melanogaster* (Supplementary Table 8)⁴⁻⁸. Among the 24,462 uORFs that met our parameter settings (Methods),
84 (0.34%) had peptides detected in at least one sample (Supplementary Table 9). Interestingly, the BLS
analysis revealed that the MS-supported uORFs present more conserved coding regions than the other uORFs
(Fig. 5e), suggesting these MS-supported uORF peptides might be functionally important. Collectively, our
results support the notion that most uORFs play regulatory roles and their start codons are maintained due to
functional constraints, and only a tiny fraction ($< 1\%$) of the uORFs might encode peptides that are maintained
by natural selection during evolution.”

In Discussion (Page 13, Lines 438-442), we also revisit this point with the following sentences “Overall,
our results suggest that the major function of uORFs is to fine-tune CDS translation rather than to encode
conserved peptides. Nevertheless, we do not deny that some uORFs can encode functional peptides, as clearly
demonstrated by previous studies⁹⁻¹¹. Of note, both our PhyloCSF analyses and MS data analyses suggest that
a small fraction ($< 1\%$) of uORFs might produce peptides.”

3) About uORF definition. In this study, uORF is defined as a 5' UTR region starting with ATG and ending
with an in-frame stop codon (TAG, TAA or TGA). The uORF definition is problematic. First, they overlooked
uORFs starting with non-canonical start codons such as CTG, TTG or ATT. Previous studies have suggested
that non-canonical start codons are more prevalence than canonical start codon (i.e. ATG) in uORFs. Second, I
would define these regions as putative uORFs or potential uORFs, because majority of these so-called “uORFs”
are not translatable. Only a very small number of these putative uORFs are real uORFs with significant protein

translation. Analysis based on these putative uORFs will be strongly affected by huge amount of false positives
(or background noises), and can not be used to support the conclusions on uORFs.

**Response:** Thank you for raising this concern. We apologize for not clearly stating that the focus of this study
was canonical uORFs that start with AUG. As most species surveyed in this study currently have no ribosome
profiling data, and it is very challenging to predict the noncanonical uORFs *in silico* reliably, we only focused
on the putative canonical uORFs which start with the AUG start codon. In this revised manuscript, we have
inserted the following sentences (Lines 73-77): “As most species surveyed in this study currently have no
ribosome profiling data, and it is very challenging to predict the noncanonical uORFs *in silico* reliably, we only
focused on the putative canonical uORFs which start with the AUG start codon. Hence, in what follows, the
uORFs analyzed in this study are restricted to the putative canonical uORFs unless explicitly stated otherwise
(all the annotated uORFs are presented in figshare²⁶).”

For the putative canonical uORFs, recent studies suggest that most of them showed evidence of translation,
although the signal of translation is dependent on the sequencing coverage of Ribo-Seq and the number of
samples surveyed in a species. For example, based on the currently available ribosome profiling data from
humans, mice, and flies, we found that approximately 70-90% of canonical uORFs can be translated
(Supplementary Table 3). Moreover, our previous analysis suggested that many uORFs are selectively used
during development or in different tissues¹⁹, which suggested that more translated uORFs can be found if we
profile more developmental stages, tissues, or cell lines from a species. Therefore, by focusing on the putative
canonical uORFs only, we can limit the influence of potential false positives.

We agree with the reviewer that only a small number of noncanonical uORFs might be real, considering
the large number of putative non-AUG uORFs in the genomes (approximately 1.2 million in humans, as reported
by McGillivray *et al.*). In the revised manuscript, we have added new analyses regarding the biological functions
of noncanonical uORFs from two aspects. First, we extracted previously published functional and population
genomic data and examined whether variations in uORF start codons influence the translation efficiency of the
main CDSs among different samples (Fig. 7a). Among the potentially functional uORFs in humans predicted
by McGillivray *et al.*¹⁴, 146 canonical and 796 noncanonical uORFs had genetic variants in their start codons
among these samples (only variants with minor allele frequency $\geq 5\%$ were considered in the analysis). We
performed linear regressions to assess the regulatory impact of uORF alteration on the translation of down-
stream CDSs, with a positive slope value in the regression meaning that the presence of a uORF in certain
individuals is associated with a decrease in the translation efficiency of the downstream CDS in those
individuals, and vice versa. A general trend was the slope values were overall positive for the canonical uORFs,
while the slope values for the noncanonical uORFs fluctuated around 0 (Fig. 6b). This comparison suggests that
in human populations, the noncanonical uORFs overall have relatively limited repressive effects on CDS
translation compared to the canonical uORFs.

Next, we experimentally verified the influence of both types of uORFs on CDS translation, we sampled 80
human uORFs and performed luciferase reporter assays in HEK293FT cells (Supplementary Fig. 17). These
tested uORFs, which included 42 canonical and 38 noncanonical ones, were predicted potentially functional by
McGillivray *et al.*¹⁴ and had polymorphic start codons in human populations. For each uORF, we compared the

repressive effect of the annotated uORF allele versus that of the non-uORF allele in suppressing the translation
of the reporter gene. Although occasionally the non-uORF allele had a stronger repressive effect than the uORF
allele, the general trend was that the uORF allele had a stronger effect than the non-uORF allele in suppressing
translation (Fig. 6c-d). Moreover, a significantly higher proportion of the canonical (55%, 23/42) than the
noncanonical (26%, 10/38) uORFs exhibited the pattern that the annotated uORF allele showed a significantly
stronger repressive effect on the CDS translation than the no-uORF allele ($P = 0.013$, Fisher's exact test, Fig.
6c-d). Also, the difference in CDS translation suppression between the uORF and the non-uORF allele is
significantly larger for the canonical than the noncanonical uORFs ($P = 0.006$, WRST). Altogether, these results
reinforced the thesis that the noncanonical uORFs overall have weaker repressive effects on CDS translation
than the canonical uORFs. We fully described these new results in the Section “**Comparing the canonical
versus noncanonical uORFs in repressing CDS translation in human populations**” (Page 14, Lines 384-
412). The new results are described as follows:

“To test whether the noncanonical uORFs influence the translation of CDSs, we extracted high-quality
genotyping, mRNA-Seq, and Ribo-Seq data of 60 human lymphoblastoid cell lines from previous studies^{27,28},
and examined whether variations in uORF start codons influence the translation efficiency of the main CDSs
among different samples (Fig. 7a). Among the potentially functional uORFs in humans predicted by
McGillivray *et al.*¹⁴, 146 canonical and 796 noncanonical uORFs had genetic variants in their start codons
among these samples (only variants with minor allele frequency $\geq 5\%$ were considered in the analysis). We
performed linear regressions to assess the regulatory impact of uORF alteration on the translation of down-
stream CDSs, with a positive slope value in the regression meaning that the presence of a uORF in certain
individuals is associated with a decrease in the translation efficiency of the downstream CDS in those
individuals, and vice versa (Methods). A general trend was the slope values were overall positive for the
canonical uORFs, while the slope values for the noncanonical uORFs fluctuated around 0 (Fig. 6b). This
comparison suggests that in human populations, the noncanonical uORFs overall have relatively limited
repressive effects on CDS translation compared to the canonical uORFs, although we cannot exclude the
possibility that a small fraction of the noncanonical uORFs might have strong repressive effects on the
translation of downstream CDSs.

To experimentally verify the influence of both types of uORFs on CDS translation, we sampled 80 human
uORFs and performed luciferase reporter assays in HEK293FT cells (Supplementary Fig. 17). These tested
uORFs, which included 42 canonical and 38 noncanonical ones, were predicted potentially functional by
McGillivray *et al.*¹⁴ and had polymorphic start codons in human populations. For each uORF, we compared the
repressive effect of the annotated uORF allele versus that of the non-uORF allele in suppressing translation of
the reporter gene. Although occasionally the non-uORF allele had a stronger repressive effect than the uORF
allele, the general trend was that the uORF allele had a stronger effect than the non-uORF allele in suppressing
translation (Fig. 6c-d). Moreover, a significantly higher proportion of the canonical (55%, 23/42) than the
noncanonical (26%, 10/38) uORFs exhibited the pattern that the annotated uORF allele showed a significantly
stronger repressive effect on the CDS translation than the non-uORF allele ($P = 0.013$, Fisher's exact test, Fig.
6c-d). Also, the difference in CDS translation suppression between the uORF and the non-uORF allele is

significantly larger for the canonical than the noncanonical uORFs ($P = 0.006$, WRST). Altogether, these results
reinforced the thesis that the noncanonical uORFs overall have weaker repressive effects on CDS translation
than the canonical uORFs.”

For example,

i) they found O/E ratio (based on these putative uORFs) is significant lower than 1, suggesting that “purifying
selection is the major force shaping the prevalence of uORFs”. This result only suggests that ATG triplets are
depleted in 5' UTR. Purifying selection for ATG triplets in 5' UTR does not mean a necessary of selection for
uORFs. In fact, at least in yeast, a previous study (PMC5793785) reported an elevated non-canonical start codon
in 5' UTR, indicating a possibility to maintain some kinds of uORF translation.

Response: Thank you for pointing this out. We apologize that in our previous submission, we did not clearly
explain how natural selection, i.e., positive and purifying selection, coupled with differences in gene expression
level and N_e , influence the genome-wide distribution and contents of uORFs in eukaryotes. As explained above,
in our analysis, we mainly focused on the putative canonical uORFs. In 475 out of 481 species we analyzed
(216 multicellular plants and animals, and 242 fungi and 23 protists that were added in the revised manuscript),
the O/E ratio of putative uORFs was significantly lower than 1, suggesting the prevalence of canonical uORFs
in a species was generally under purifying selection. However, we also found that positive selection can drive
the fixation of new uORFs that are beneficial in primates that typically have a small N_e . These results suggest
that positive selection might play a more important role in driving uORF evolution than previously anticipated.
Furthermore, we demonstrated that how the effective population size of a species affects the efficacy of natural
selection on the prevalence of uORFs. The whole section is presented in Lines 202-239 of Pages 7-8.

Moreover, in this revised manuscript, we further explored how gene expression level is an important factor
influencing the distribution of uORFs across genes. Our gene ontology analysis revealed that uORFs are biased
in genes of different functional categories, which are associated with gene expression levels. We also found that
purifying selection has removed deleterious uORFs in the highly expressed genes more efficiently than in the
lowly expressed genes. We presented the new analysis in the section “**Gene expression level as an important**
**factor influencing the genome-wide distributions of uORFs across genes**” (Page 5, Lines 139-167). We also
showed that the efficacy of positive selection on uORFs is stronger in highly expressed genes, and this pattern
was observed in both primates and *Drosophila* (Page 7, Lines 220-223).

Also, in Discussion, we reconciled the roles of natural selection and gene expression level on the
distribution of uORFs across species and across genes. The relevant sentences are reproduced as follows (Page
13, Lines 416-431):

“Although the prevalence of canonical uORFs in a species was generally under purifying selection, we
still found a fraction of new canonical uORFs might be favored by positive selection even in primates that
typically have a small N_e . These observations are consistent with the evolution model of uORFs we previously
proposed^{19,29}. Under that model, the majority of newly formed uORFs are deleterious and quickly removed from
the population, and a relatively smaller fraction of the new uORFs are beneficial and rapidly fixed in populations
under positive selection. After fixation, the functional uORFs, particularly the start codons, are maintained by

natural selection during evolution. Hence, although in a species the occurrence of a uORF is influenced by
positive or purifying selection, the opposing effects of positive selection and purifying selection acting on new
uORFs result in a pattern that uORFs are overall depleted in 5' UTRs. As shown in our population genetic
modeling, the efficacies of both positive and purifying selection on uORF fixation in a species are influenced
by the effective population size. Moreover, we also found that gene expression level affects the efficiency of
natural selection acting on uORF occurrences. Thus, our results have systematically demonstrated how positive
and purifying selection, coupled with differences in gene expression level and N_e , influence the genome-wide
distribution and contents of uORFs in eukaryotes. Together, our analyses provide an unprecedented overview
of the general principles underlying the distribution and sequence evolution of uORFs in eukaryotes.”

ii) The authors found that the dN/dS ratio for uORF CDS is “roughly equal to 1 between human and macaque”.
They concluded that this result supports neutral evolution of uORFs. However, because majority of “uORFs”
in their datasets are non-translatable (or not real uORF), these negative uORFs may significantly increase the
dN/dS ratio, since they encode nothing. Again, in *Drosophila*, the dN/dS ratio for all uORF CDSs is close to 1,
but later, they found that “uORFs with higher Kozak scores presented significantly lower dN/dS ratio in
*Drosophila*, suggesting a scenario in which the coding regions of uORFs with optimal Kozak sequence context
are under stronger purifying selection in *Drosophila*”. Since ATG surrounded by Kozak sequences are more
likely to be translated, I believe their negative result (i.e. dN/dS is close to 1) is due to too many negative uORFs
in their datasets.

**Response:** Thank you for raising these concerns. We apologize that in our previous submission, we might have
performed the analyses in an unnecessarily complicated approach so that our results might have been misleading
to this reviewer. In this study, we mainly focused on the putative canonical uORFs, and 69% and 89% of such
uORFs exhibit evidence of translation in humans and flies, respectively. Therefore, our results are not likely
caused by too many negative uORFs as this reviewer thought. Furthermore, we also repeated all the analyses in
this section using the uORFs that showed evidence of translation (Supplementary Fig. 11), and our conclusions
were not affected. In this new submission, we updated our analyses to avoid potential misunderstandings.

Moreover, in this revised version, we performed two additional analyses to address this reviewer’s
concerns. First, we performed PhyloCSF analysis of the coding regions of uORFs. The PhyloCSF algorithm
predicts whether a genomic region potentially represents a conserved protein-coding region or not based on
multiple sequence alignments³, and a positive PhyloCSF score means that region is more likely to encode a
peptide. As negative controls, we also calculated the PhyloCSF scores for 20,000 randomly selected ORFs in 3’
UTRs (downstream ORFs, dORFs), as these dORFs have little chance of translation. By comparing the
PhyloCSF scores of the uORFs that showed evidence of translation with the ribosome profiling data versus the
random dORFs, we estimated that in humans, 0.44% (161 out of 36,655) of the translated uORFs showed
evidence of encoding conserved peptides, and that in *Drosophila*, 0.80% (102 of 12,754) of the translated
uORFs might encode conserved peptides. Overall, these analyses suggest that less than 1% canonical uORFs
might encode conserved peptides.

Next, we examined public mass spectrometry (MS) datasets for evidence of uORF-encoded peptides in
*Drosophila*. We analyzed the mass spectrometry (MS) data from 38 samples of different developmental stages
or tissues of *D. melanogaster* from previous studies⁴⁻⁸ (see Supplementary Table 8 for details). Among the
24,462 uORFs that met our parameter settings, 84 (0.34%) had peptides detected in at least one sample (see
Supplementary Table 9 for details). In combination with our finding that most uORFs do not encode conserved
peptides, these results suggest that only a very small fraction ($< 1\%$) of the uORFs might encode peptides that
are maintained by natural selection during evolution. Therefore, we obtained consistent results among our
molecular evolution and population genetic analysis, the phyloCSF analysis, and the MS data re-analysis. The
new results are described in Lines 318-338 of Page 10.

iii) the same problem can be found in the analysis of “evolution of contextual characteristics that influence
uORF translation”.

**Response:** The effect of potential negative uORFs should be limited since we only focused on canonical uORFs,
most of which are evidenced of translation with the ribosome profiling data in model organisms.

4) Page 4, line 94. “gene expression level is a major determinant of the uORF distribution across genes in a
eukaryotic species” Because the number of putative uORFs positively correlates with 5' UTR length, I
wondered whether 5' UTR may confound the correlation of putative uORF number to gene expression.

**Response:** Thank you for this insightful comment. To address this concern, in this revised version, we grouped
genes of a species into bins of equal size based on their expression level and calculated the O/E ratio of uORFs
for genes in each bin. As a result, the potential confounding effect of differences in the 5' UTR length is also
properly controlled. In all the five species we examined, the O/E ratio was lower than 1 in each bin
(Supplementary Fig. 5c), suggesting that purifying selection was the dominant evolutionary force acting on the
uORF occurrence regardless of gene expression levels. Interestingly, we observed significant anticorrelations
between the gene expression level and O/E ratio of uORFs in each species, suggesting that purifying selection
acting on uAUGs is relatively weak for lowly expressed mRNAs. The new analysis is described as follows
(Page 6, Lines 152-167):

“Since gene expression level affects the efficacy of natural selection²², we further asked whether the
efficacy of purifying selection is reduced in removing deleterious uORFs in lowly expressed genes. We grouped
genes of a species into 20 equal-sized bins based on increasing expression levels and calculated the O/E ratio
of uORFs in each bin. In all the five species we examined, the O/E ratio was lower than 1 in each bin
(Supplementary Fig. 5c), suggesting that purifying selection was the dominant evolutionary force acting on the
uORF occurrence regardless of gene expression levels. Interestingly, we observed significant anticorrelations
between the gene expression level and O/E ratio of uORFs in each species, suggesting that purifying selection
acting on uAUGs is relatively weak for lowly expressed mRNAs.

Thus, our results suggest that gene expression level is an important factor influencing uORF distribution
across genes in a eukaryotic species. It is possible that excessive uORFs in highly expressed genes might cause
insufficient protein output, which is harmful to the organisms. We postulate that purifying selection has removed

deleterious uORFs in the highly expressed genes more efficiently than in the lowly expressed genes. On the
other hand, genes in certain functional categories, such as transcriptional factors, which are likely to be lowly
expressed, might be preferentially suppressed by uORFs at the translational level for optimizing protein
production. Further studies are needed to investigate the relative importance of the two mechanisms in shaping
the anticorrelation between gene expression level and uORF occurrence.”

5) Page 4, line 101, “while maintaining the same dinucleotide frequency”. Please explain why dinucleotide
frequency is maintained. Does single, or trip-nucleotide frequency significantly affect O/E ratio?

**Response:** Thank you for the helpful suggestion. In this revision, we further explained the reason for
maintaining the same dinucleotide frequency as follows (Page 4, Lines 107-110): “We maintained the same
dinucleotide frequencies in each sequence during shuffling for two reasons. First, the stacking energy of a new
base pair is influenced by the neighboring base pairs in an RNA molecule^{30,31}. Second, the biased mutations in
certain dinucleotide contexts, such as from CpG to TpG mutations in mammals, might also affect the prevalence
of uORFs.”

Since the AUG start codon has three nucleotides, maintaining the trinucleotide frequency means that the
frequency of every triplet in a sequence is unchanged in the shuffled sequence. Therefore, the O/E ratio for any
triplet (including ATG) will be 1, which is not appropriate for the current study.

6) O/E ratio in 5' UTR might be ok to estimate the selection for ATG triplets. To strength the results, O/E ratio
in 3' UTR should be considered as negative control, since translation in 3' UTR ORFs is less likely than that in
5' UTR. In addition, it would be great if O/E ratios for the other 61 triplets are displayed.

**Response:** Thank you for these helpful suggestions. In this revision, we followed these suggestions and
calculated the O/E ratios for all triplets in the 5' UTR and 3' UTR. The new results have been included in the
revised manuscript, and relevant sentences read as follows (Page 4, Lines 113-117):

“Since AUG is the defining feature of a canonical uORF, the O/E ratio is essentially the observed/expected
number of AUG triplets in the 5' UTRs. As a negative control, we also calculated the O/E ratio of all the other
63 possible triplets in 5' UTRs and 3' UTRs separately in each species. Of note, AUG had the lowest relative
O/E ratio (5' UTRs over 3' UTRs) among all the 64 possible triplets (Supplementary Fig. 2), supporting the
notion that purifying selection is the major force shaping the prevalence of uORFs in the eukaryotic genomes.”

7) Page 5, line 122. “The O/E ratio varied wildly across the 216 species”, is this ratio affected by different
background ATG frequency (E) across the species?

**Response:** Thank you for asking this question. Overall, the background ATG frequency (E) has little influence
on the O/E ratio of ATG in the 5' UTRs among different species. E is highly correlated with the observed
frequency of ATG (O) in a species (Fig. R1 left). In contrast, the O/E ratio showed a much weaker correlation
with the background ATG frequency (E), which is likely due to the uneven distribution of background ATG
frequencies across species. Moreover, the O/E ratio enabled the efficient measurement of selective pressure on
uORF depletion in a given species, as shown in previous studies^{19,32-35}.

Fig. R1. Influence of the expected frequency of ATG. Left panel, the relationship between the expected
 frequency of ATG and the O/E ratio among different species. Right panel, the relationship between the expected
 frequency of ATG and the observed frequency of ATG. Spearman's correlation analysis was performed for
 each plot.

8) Page 8, they found longer uORFs have fewer conserved peptides. This is a little unexpected to me. Because
 uORF translation is energy-consuming. If a uORF plays regulator role, a shorter ORF is sufficient to block
 ribosome scanning to downstream region. The longer ORF does not significant benefit the regulator role, but
 indeed consume more energy.

**Response:** Thank you for the suggestion. We have revised the relevant text to reflect this point (Page 10, Lines
 291-296):

"Of note, a strong anticorrelation was observed between the BLSs and the lengths of uORF peptides in
 both humans and flies (see Fig. 4c and 4d), suggesting the peptides encoded by long uORFs are less likely to
 be maintained during evolution because they were more likely disrupted by stop codons or frameshifts. Also, if
 the major function of uORFs is to regulate CDS translation, a longer uORF might be less advantageous than a
 shorter one because the translation of a longer uORF consumes more energy and metabolites, which might be
 harmful to the host organisms."

**References:**

Sengupta, S. & Higgs, P. G. Pathways of Genetic Code Evolution in Ancient and Modern Organisms. *J*
 *Mol Evol* **80**, 229-243, doi:10.1007/s00239-015-9686-8 (2015).
 Swart, E. C., Serra, V., Petroni, G. & Nowacki, M. Genetic Codes with No Dedicated Stop Codon:
 Context-Dependent Translation Termination. *Cell* **166**, 691-702, doi:10.1016/j.cell.2016.06.020
 (2016).
 Lin, M. F., Jungreis, I. & Kellis, M. PhyloCSF: a comparative genomics method to distinguish protein
 coding and non-coding regions. *Bioinformatics* **27**, i275-282, doi:10.1093/bioinformatics/btr209
 (2011).
 Xing, X. *et al.* Qualitative and quantitative analysis of the adult *Drosophila melanogaster* proteome.
 *Proteomics* **14**, 286-290, doi:10.1002/pmic.201300121 (2014).
 Casas-Vila, N. *et al.* The developmental proteome of *Drosophila melanogaster*. *Genome Res* **27**, 1273-

1285, doi:10.1101/gr.213694.116 (2017).

Ashley, J. *et al.* Retrovirus-like Gag Protein Arc1 Binds RNA and Traffics across Synaptic Boutons. *Cell* **172**, 262-274 e211, doi:10.1016/j.cell.2017.12.022 (2018).

Kuznetsova, K. G. *et al.* Proteogenomics of Adenosine-to-Inosine RNA Editing in the Fruit Fly. *J Proteome Res* **17**, 3889-3903, doi:10.1021/acs.jproteome.8b00553 (2018).

Sabbadin, F. *et al.* An ancient family of lytic polysaccharide monooxygenases with roles in arthropod development and biomass digestion. *Nat Commun* **9**, 756, doi:10.1038/s41467-018-03142-x (2018).

Mackowiak, S. D. *et al.* Extensive identification and analysis of conserved small ORFs in animals. *Genome Biol* **16**, 179, doi:10.1186/s13059-015-0742-x (2015).

Aspden, J. L. *et al.* Extensive translation of small Open Reading Frames revealed by Poly-Ribo-Seq. *eLife* **3**, e03528, doi:10.7554/eLife.03528 (2014).

van der Horst, S., Snel, B., Hanson, J. & Smeekens, S. Novel pipeline identifies new upstream ORFs and non-AUG initiating main ORFs with conserved amino acid sequences in the 5' leader of mRNAs in *Arabidopsis thaliana*. *Rna* **25**, 292-304, doi:10.1261/rna.067983.118 (2019).

Alexa, A. & Rahnenfuhrer, J. topGO: enrichment analysis for gene ontology. *R package* (2019).

Benjamini, Y. & Hochberg, Y. Controlling the false discovery rate: a practical and powerful approach to multiple testing. *Journal of the Royal statistical society: series B (Methodological)* **57**, 289-300 (1995).

McGillivray, P. *et al.* A comprehensive catalog of predicted functional upstream open reading frames in humans. *Nucleic acids research* **46**, 3326-3338, doi:10.1093/nar/gky188 (2018).

Xu, G. *et al.* uORF-mediated translation allows engineered plant disease resistance without fitness costs. *Nature* **545**, 491-494, doi:10.1038/nature22372 (2017).

Zhang, H. *et al.* Genome editing of upstream open reading frames enables translational control in plants. *Nature Biotechnology* **36**, 894-898, doi:10.1038/nbt.4202 (2018).

Ferreira, J. P., Overton, K. W. & Wang, C. L. Tuning gene expression with synthetic upstream open reading frames. *Proceedings of the National Academy of Sciences* **110**, 11284, doi:10.1073/pnas.1305590110 (2013).

Ye, Y. *et al.* Analysis of human upstream open reading frames and impact on gene expression. *Hum Genet* **134**, 605-612, doi:10.1007/s00439-015-1544-7 (2015).

Zhang, H. *et al.* Genome-wide maps of ribosomal occupancy provide insights into adaptive evolution and regulatory roles of uORFs during *Drosophila* development. *PLoS Biol* **16**, e2003903, doi:10.1371/journal.pbio.2003903 (2018).

Michel, A. M., Kiniry, S. J., O'Connor, Patrick B F., Mullan, J. P. & Baranov, P. V. GWIPS-viz: 2018 update. *Nucleic Acids Research* **46**, D823-D830, doi:10.1093/nar/gkx790 (2017).

Eisenberg, E. & Levanon, E. Y. Human housekeeping genes, revisited. *Trends in Genetics* **29**, 569-574, doi:10.1016/j.tig.2013.05.010 (2013).

Zhang, J. & Yang, J.-R. Determinants of the rate of protein sequence evolution. *Nature reviews. Genetics* **16**, 409-420, doi:10.1038/nrg3950 (2015).

Gonzalez-Perez, A., Sabarinathan, R. & Lopez-Bigas, N. Local Determinants of the Mutational Landscape of the Human Genome. *Cell* **177**, 101-114, doi:10.1016/j.cell.2019.02.051 (2019).

Kitano, S., Kurasawa, H. & Aizawa, Y. Transposable elements shape the human proteome landscape via formation of cis-acting upstream open reading frames. *Genes Cells* **23**, 274-284, doi:10.1111/gtc.12567 (2018).

Kimura, M. Diffusion models in population genetics. *Journal of Applied Probability* (1964).

Hong, Z. *et al.* The annotation of uORFs in 481 eukaryotes. *figshare*, doi:10.6084/m9.figshare.9980441.v2 (2020).

Auton, A. *et al.* A global reference for human genetic variation. *Nature* **526**, 68-74, doi:10.1038/nature15393 (2015).

Battle, A. *et al.* Genomic variation. Impact of regulatory variation from RNA to protein. *Science* **347**, 664-667, doi:10.1126/science.1260793 (2015).

Zhang, H., Wang, Y. & Lu, J. Function and Evolution of Upstream ORFs in Eukaryotes. *Trends in Biochemical Sciences* **44**, 782-794, doi:10.1016/j.tibs.2019.03.002 (2019).

Clote, P., Ferré, F., Kranakis, E. & Krizanc, D. Structural RNA has lower folding energy than random RNA of the same dinucleotide frequency. *RNA (New York, N.Y.)* **11**, 578-591, doi:10.1261/rna.7220505 (2005).

Workman, C. & Krogh, A. No evidence that mRNAs have lower folding free energies than random sequences with the same dinucleotide distribution. *Nucleic acids research* **27**, 4816-4822,

doi:10.1093/nar/27.24.4816 (1999).
Kozak, M. Possible role of flanking nucleotides in recognition of the AUG initiator codon by eukaryotic
ribosomes. *Nucleic Acids Res* **9**, 5233-5252 (1981).
Lynch, M., Scofield, D. G. & Hong, X. The evolution of transcription-initiation sites. *Mol Biol Evol* **22**,
1137-1146, doi:10.1093/molbev/msi100 (2005).
Neafsey, D. E. & Galagan, J. E. Dual Modes of Natural Selection on Upstream Open Reading Frames.
*Molecular Biology and Evolution* **24**, 1744-1751, doi:10.1093/molbev/msm093 (2007).
Rogozin, I. B., Kochetov, A. V., Kondrashov, F. A., Koonin, E. V. & Milanesi, L. Presence of ATG
triplets in 5' untranslated regions of eukaryotic cDNAs correlates with a 'weak' context of the start
codon. *Bioinformatics* **17**, 890-900 (2001).

Reviewers' Comments:

Reviewer #1:

Remarks to the Author:

The authors performed several analyses which expanded the manuscript substantially and well beyond my expectations. Because of such an extensive revision, a few issues emerged that are specific for the revised version, which I mention below. However, the significance of these issues is relatively minor. I think that the general message of this manuscript is adequately supported.

Comments pertinent for the revised version.

My very first comment related to the apparent impossibility of uORF (if we define uORFs as short translated sequences upstream of CDS) existence in some protists where genetic codes do not allow translation termination of translation in the internal positions of mRNA.

In these organisms any translation initiation event would result in a production of a long protein. So I suggested that the authors should simply acknowledge this fact and modify their title and discussion to make it more specific, i.e. plants and animals, but not eukaryotes in general.

Instead of following this simple suggestion, the authors chose a hard way, they decided to expand their analysis to include fungi and protists. Of course, this broadens the manuscript and I agree that the use of eukaryotes in the title is appropriate now. However, it made the work not only broader, but also more complicated. While the situation with fungi is similar to that of plants and animals, protists seem to be different and I would like to ask authors to make additional changes in their manuscript to clarify the situation.

First, protists are hugely diverse phylogenetically and exhibit considerable diversity in organisation of their genetic information in their genomes and probably in mRNAs as well. Because of that the analysis of 23 species may not be adequate to obtain a general picture.

Second, the analysis was done incorrectly. The authors did not take into account the diversity of their genetic codes in these species: "Putative uORFs that start with AUG codons and end with stop codons (UAA/UAG/UGA) were identified from the annotated 5' UTRs of protein-coding genes". This is clearly wrong for the organisms that do not use one or several of these codons as stops. However, this mistake, as far as I understand is not very critical because in my understanding of the authors analyses the most important part is the location of ATGs and where uORFs end affects only their classification as overlapping. Note, that all uORFs in *C. magnum* and other species with no internal stops are oORFs.

Relevant to this is the definition of uORFs provided in the revised version with which I cannot fully agree: "Based on the position and frame of a uORF relative to the downstream CDS, uORFs can be classified into nonoverlapping uORFs, out-of-frame overlapping uORFs (oORFs), and N-terminal extensions"

I strongly disagree with referring to N-terminal extensions as uORFs. If there is an upstream AUG that is in-frame with annotated AUG and initiation takes place on it, we are dealing with misannotated CDS, rather than a distinct phenomenon such as translation of short ORFs occurring upstream of CDS. More than one start could be used for initiation at protein coding ORFs (see our recent work related to this phenomenon in human mRNAs - Benitez-Cantos et al 2020 <https://doi.org/10.1101/gr.257352.119>) and we could refer to the products of these alternative initiation events as truncations or extensions relative to each other, but certainly the longer extended proteoform is not an uORF, it is CDS.

The authors noted that O/E ratio in *C. magnum* is not below 1. Could this be simply because what the authors refer to as uORFs are either CDS starts or internal methionine codons? Hence there is no selection against them? But if it is the case, we get back to the thesis that I stated earlier –

there are no uORFs (in classical sense) in the species with no internal stops.

It seems to me that protists are a special case and their comprehensive analysis could be difficult and inappropriate for this manuscript which is already quite substantial. To some extent the authors already acknowledge this by saying "how the genetic code reassignments affect the distribution and evolution of uORFs in certain protists deserves further study." I think this should be stated more clearly, that the occurrence of uORFs in some protists with genetic code variants may differ substantially from that of most other eukaryotic organisms.

Finally, in relation to the discovery of "stopless" genetic code in *C. magnum* the authors cite one work, however, this discovery was made in two laboratories independently at the same time so both references should be used:

1. Swart et al 2016 <https://doi.org/10.1016/j.cell.2016.06.020>

2. Heaphy et al 2016 <https://doi.org/10.1093/molbev/msw166>

Perhaps the authors may wish to mention other species with the genetic codes which are incompatible with uORFs existence, though this is not necessary unless the authors wish to discuss it:

Blastocrithidia, see Zahonova et al 2016 <https://doi.org/10.1016/j.cub.2016.06.064>)

Ciliate Euplotes, see Lobanov et al 2017 <https://doi.org/10.1038/nsmb.33300>

Amoebophrya sp. ex *Karlodinium veneficum*, see Bachvaroff 2019 <https://doi.org/10.1371/journal.pone.0212912>.

//Pasha Baranov//

Reviewer #2:

Remarks to the Author:

The authors have addressed all my comments. The new data and discussions on canonical and non-noncanonical are very interesting. I would be happy to see this work published.

If I may I want to bring up a few minor points:

1. Figure 1a, middle panel. Almost all species except for human, mouse and fruit fly contains a large fraction of mRNAs without 5' UTR. It is quite unusual, since majority of translation rely on a scanning process in 5' UTR to start translation. Is this due to a lack of accurate annotations on 5' UTR?

2. Supplementary Figure 2. It is interesting that the ATG-like triplets (e.g. ATT, TTG, ATC, GTG) are over-represented in 5' UTR, compared with 3' UTR. Does that indicate that non-canonical uORFs are more preferred within 5' UTR? Or alternatively, is this a consequence of depletion of canonical uORFs, since a single mutation on ATG triplets can easily lead to a switch of canonical uORFs to non-canonical uORFs (ATG -> TTG). As shown in this study, non-canonical uORFs are less effective to repress CDS translation, therefore, these single mutations are not removed during evolution.

3. Rebuttal letter, line 466. "we found that approximately 70-90% of canonical uORFs can be translated (Supplementary Table 3)." It is an interesting data, but I can not find this dataset in Supp. Table 3, or am I missing something? By the way, how to define a translatable uORF, by

using Ribo-seq or TIS-seq. Was that done based on a threshold or by using other ORF prediction tools (e.g. PMID: 26657557)?

Thanks for the opportunity to review this article

Point-to-point response

Reviewer #1 (Remarks to the Author)

The authors performed several analyses which expanded the manuscript substantially and well beyond my expectations. Because of such an extensive revision, a few issues emerged that are specific for the revised version, which I mention below. However, the significance of these issues is relatively minor. I think that the general message of this manuscript is adequately supported.

Response: Thank you for the thorough review of our manuscript. We appreciate the positive feedbacks from this reviewer. We have addressed your comments and concerns in this revised version. Please refer to the point-to-point responses for details.

Comments pertinent for the revised version.

My very first comment related to the apparent impossibility of uORF (if we define uORFs as short translated sequences upstream of CDS) existence in some protists where genetic codes do not allow translation termination of translation in the internal positions of mRNA. In these organisms any translation initiation event would result in a production of a long protein. So I suggested that the authors should simply acknowledge this fact and modify their title and discussion to make it more specific, i.e. plants and animals, but not eukaryotes in general. Instead of following this simple suggestion, the authors chose a hard way, they decided to expand their analysis to include fungi and protists. Of course, this broadens the manuscript and I agree that the use of eukaryotes in the title is appropriate now. However, it made the work not only broader, but also more complicated. While the situation with fungi is similar to that of plants and animals, protists seem to be different and I would like to ask authors to make additional changes in their manuscript to clarify the situation.

Response: Thanks for the comments and suggestions. The major critique raised by this reviewer is that in our previously submitted version, three of the 23 protists do not use the standard genetic code, which makes the identification of uORFs questionable in these three species.

In this revised version, we only considered the 20 protists that use the standard genetic code, and excluded *Condylostoma magnum* and *Parduczia* sp., both of which had no dedicated stop codons (Heaphy et al., 2016; Swart et al., 2016), and *Ichthyophthirius multifiliis*, in which UAA and UAG are reassigned to encode glutamine (Coyne et al., 2011).

We inserted the sentence “Since many protists use alternative nuclear genetic codes involving stop-codon reassignments⁶⁸⁻⁷³ or obligatory frameshifting at internal stop codons⁷⁴, here we only focused on 20 protists that use the standard genetic code (Supplementary Table 1).” in the revised manuscript to clarify this point (Lines 87- 89, Page 4).

First, protists are hugely diverse phylogenetically and exhibit considerable diversity in organisation of their genetic information in their genomes and probably in mRNAs as well. Because of that the analysis of 23 species may not be adequate to obtain a general picture.

Response: Thanks for raising this concern. We emphasized this point in the Discussion with the following sentences (Lines 475-481, Page 15):

“Protists have a very high phylogenetic diversity¹²⁸, and many protists use alternative nuclear genetic codes involving stop-codon reassignments^{68,69} and obligatory frameshifting at internal stop codons⁷⁴. In protists with no dedicated stop codons⁷¹, such as *Condylostoma magnum*^{70,71}, *Parduczia* sp.⁷¹, *Blastocrithidia*⁷², and *Amoebophrya* sp. ex *Karlodinium veneficum*⁷³, translation from any possible uAUG is supposed to terminate

near the end of a transcript and overlaps with the main CDS, which results in a different protein. Thus, the occurrence of uORFs in protists with alternative genetic decoding schemes might differ considerably from that of most other eukaryotes. In this study, we only focused on 20 protists that use the standard genetic code.”

Second, the analysis was done incorrectly. The authors did not take into account the diversity of their genetic codes in these species: “Putative uORFs that start with AUG codons and end with stop codons (UAA/UAG/UGA) were identified from the annotated 5' UTRs of protein-coding genes”. This is clearly wrong for the organisms that do not use one or several of these codons as stops. However, this mistake, as far as I understand is not very critical because in my understanding of the authors analyses the most important part is the location of ATGs and where uORFs end affects only their classification as overlapping. Note, that all uORFs in *C. magnum* and other species with no internal stops are oORFs.

Response: Thanks for pointing this out. In this revised version, we corrected this mistake by only focusing on 20 protists that use the standard genetic code. Please refer to our response above.

Relevant to this is the definition of uORFs provided in the revised version with which I cannot fully agree: “Based on the position and frame of a uORF relative to the downstream CDS, uORFs can be classified into nonoverlapping uORFs, out-of-frame overlapping uORFs (oORFs), and N-terminal extensions”

I strongly disagree with referring to N-terminal extensions as uORFs. If there is an upstream AUG that is in-frame with annotated AUG and initiation takes place on it, we are dealing with misannotated CDS, rather than a distinct phenomenon such as translation of short ORFs occurring upstream of CDS. More than one start could be used for initiation at protein coding ORFs (see our recent work related to this phenomenon in human mRNAs - Benitez-Cantos et al 2020 <https://doi.org/10.1101/gr.257352.119>) and we could refer to the products of these alternative initiation events as truncations or extensions relative to each other, but certainly the longer extended proteoform is not an uORF, it is CDS.

Response: Thank you very much for pointing this out. The nomenclature of ORFs has been inconsistent among different studies, as discussed in a recent review (Orr et al., 2019). For an ORF that has the AUG start codon located in the 5' UTR (defined as “uAUG” in our study), it can function as the start codon of a uORF that has a stop codon either preceding the start codon of the downstream CDS (nonoverlapping uORF, nORF) or residing in the body of the downstream CDS (out-of-frame overlapping uORF, oORF). Also, an uAUG can function as the start codon of an ORF whose stop codon overlaps with the stop codon of the downstream CDS (N-terminal extension, NTE).

Although several studies used a strict definition and only considered nORFs as uORFs (Calviello, et al. 2016; Johnstone, et al. 2016; Whiffin, et al. 2020), many studies broadly treated all the three categories of ORFs as uORFs as they only required the start codons to reside in 5' UTRs (Brar, et al. 2012; Aspden, et al. 2014; Chew, et al. 2016; McGillivray, et al. 2018; Niu, et al. 2020). Several studies also argued that only oORFs and nORFs should be treated as uORFs, and NTEs should be treated as alternative initiation of CDS (Calvo, et al. 2009; Benitez-Cantos, et al. 2020; Chen, et al. 2020).

Previously, we took the broad definition of uORFs and considered all the three categories of uAUGs in 5' UTRs as uORFs. However, as suggested by this reviewer, NTEs are alternative initiation sites of CDSs and thus might differ from nORFs and oORFs in function and sequence evolution. Therefore, in this revision, we followed the suggestions of this reviewer and thoroughly revised the manuscript. In short, we only treated nORFs and oORFs as uORFs in this analysis, although we still considered the start codons of NTEs as a type of uAUGs in this new version. Specifically, we made the following changes:

First, we defined uORFs in the Introduction of the revised manuscript with the following sentences (Line 40-45, Page 3):

“For an AUG triplet in the 5' UTR (defined as “uAUG” hereafter), it can function as the start codon of a uORF that has a stop codon either preceding the start codon of the downstream CDS (nonoverlapping uORF, nORF) or residing in the body of the downstream CDS (out-of-frame overlapping uORF, oORF)^{4,11-18}. Less frequently, an uAUG can function as the start codon of an ORF whose stop codon overlaps with the stop codon of the downstream CDS (N-terminal extension, NTE)^{4,19-21}.”

Second, we also described the compositions of uORFs and NTEs in the examined eukaryotes with the following sentences (Lines 109-112, Page 4):

“The vast majority (> 97%) of the uAUGs identified in the 478 eukaryotic species were start codons of putative canonical uORFs. Specifically, in a species, the percentage (mean \pm s.e.) of nORFs, oORFs, and NTEs was $83.45 \pm 0.41\%$, $14.24 \pm 0.34\%$, and $2.31 \pm 0.15\%$, respectively. The detailed information for the uORFs (nORFs and oORF) and NTEs is presented in Supplementary Table 1.”

Third, we focused our main analyses on the putative canonical uORFs. We emphasized this point with the following sentences (Lines 153-156, Page 6):

“Overall, these results suggest that uAUGs were selected against in 5' UTRs, and the NTEs, which only accounted for a small fraction ($\sim 2.31\%$ on average) of the uAUGs, were also shaped by strong purifying selection during evolution. Since uORFs (nORFs and oORFs) and NTEs might have different mechanisms in regulating gene expression and function, in what follows, we only focused on the putative canonical uORFs.”

Fourth, with the new definition of uORFs, we updated all the relevant analyses that were presented in the main figures (Figs. 1, 2, 3, 5, 6) and supplementary information (Supplementary Figs. 5, 7, 8, 10, 15, and 16; Supplementary Tables 2, 3, 4, 7, and 9). Figures 4 and 7 were not affected since NTEs had been excluded in the previous version.

Despite the extensive updates of the analyses, our conclusions were not affected. This is expected since the NTEs only account for a minor fraction ($2.31 \pm 0.14\%$, mean \pm s.e.) of uAUGs in the species we investigated.

We hope these changes are satisfactory to this reviewer. We are certainly willing to make further revisions if this reviewer thinks additional changes are needed.

The authors noted that O/E ratio in *C. magnum* is not below 1. Could this be simply because what the authors refer to as uORFs are either CDS starts or internal methionine codons? Hence there is no selection against them? But if it is the case, we get back to the thesis that I stated earlier – there are no uORFs (in classical sense) in the species with no internal stops.

Response: Thanks for raising this concern. In this revised version, we removed *C. magnum*, as well as *Parduezia* sp. and *Ichthyophthirius multifiliis*, from the analysis. Please refer to our response above.

However, among the 20 protists that use the standard genetic code, we still found the O/E ratio for uAUGs was close to or higher than 1 in five protists. Nevertheless, we think this observation might be an artifact caused by inaccurate 5' UTR annotations in these five species, because these five protists tended to have significantly longer 5' UTRs than the other 15 protists (Supplementary Fig. 18). Importantly, the O/E ratio of uAUGs in the 5' UTR regions that are proximal to CDS (within 100 nt or 150 nt) were significantly lower than 1 in all the five

protists (Supplementary Table 11), suggesting that uAUG occurrence in 5' UTR regions proximal to CDSs is still under purifying selection in these five protists.

We reported these observations in Discussion with the following sentences (Lines 481-492, Page 15):

“In this study, we only focused on 20 protists that use standard genetic code. Although the O/E ratio of uAUGs was significantly less than 1 in all the fungi, multi-cellular plants and animals we examined, such a pattern was observed in only 15 of the 20 protists. The O/E ratio of uAUGs was close to or higher than 1 in the remaining five protists, including *Cystoisospora suis* (1.161, 95% CI 1.154~1.169), *Toxoplasma gondii* (0.998, 95% CI 0.989~1.1.007), *Nannochloropsis gaditana* (0.997, 95% CI 0.986~1.007), and two malaria vectors *Plasmodium yoelii* (1.016, 95% CI 1.008~1.025) and *Plasmodium vivax* (0.989, 95% CI 0.975~1.004). However, these five protists tended to have significantly longer 5' UTRs than the other 15 protists (Supplementary Fig. 18), suggesting this observation might be an artifact caused by inaccurate 5' UTR annotations in these five species. Indeed, the O/E ratio of uAUGs in the 5' UTR regions that are proximal to CDS (within 100 nt or 150 nt) were significantly lower than 1 in all the five protists (Supplementary Table 11), suggesting that uAUG occurrence in 5' UTR regions proximal to CDSs is still under purifying selection in these protists.”

It seems to me that protists are a special case and their comprehensive analysis could be difficult and inappropriate for this manuscript which is already quite substantial. To some extent the authors already acknowledge this by saying “how the genetic code reassignments affect the distribution and evolution of uORFs in certain protists deserves further study.” I think this should be stated more clearly, that the occurrence of uORFs in some protists with genetic code variants may differ substantially from that of most other eukaryotic organisms.

Response: Thanks for the advice. We rephrased the sentences “Thus, the occurrence of uORFs in protists with alternative genetic decoding schemes might differ considerably from that of most other eukaryotes. In this study, we only focused on 20 protists that use standard genetic code.” to make this point clearer in Discussion (Lines 479-481, Page 15).

Finally, in relation to the discovery of “stopless” genetic code in *C. magnum* the authors cite one work, however, this discovery was made in two laboratories independently at the same time so both references should be used:

1. Swart et al 2016 <https://doi.org/10.1016/j.cell.2016.06.020>
2. Heaphy et al 2016 <https://doi.org/10.1093/molbev/msw166>

Response: Thank you for pointing this out. The citations have been updated (Refs. 70 and 71) in this revision.

Perhaps the authors may wish to mention other species with the genetic codes which are incompatible with uORFs existence, though this is not necessary unless the authors wish to discuss it:

Blastocrithidia, see Zahonova et al 2016 <https://doi.org/10.1016/j.cub.2016.06.064>) Ciliate Euplotes, see Lobanov et al 2017 <https://doi.org/10.1038/nsmb.3330> Amoebophrya sp. ex *Karlodinium veneficum*, see Bachvaroff 2019 <https://doi.org/10.1371/journal.pone.0212912>.

Response: Thank you for these suggestions. These works have been briefly mentioned in Discussion as follows (Lines 475-481, Page 15):

“Protists have a very high phylogenetic diversity¹²⁸, and many protists use alternative nuclear genetic codes involving stop-codon reassignments^{68,69} and obligatory frameshifting at internal stop codons⁷⁴. In protists with no dedicated stop codons⁷¹, such as *Condylostoma magnum*^{70,71}, *Parduczia* sp.⁷¹, *Blastocrithidia*⁷², and *Amoebophrya* sp. ex *Karlodinium veneficum*⁷³, translation from any possible uAUG is supposed to terminate near the end of a transcript and overlaps with the main CDS, which results in a different protein. Thus, the

occurrence of uORFs in protists with alternative genetic decoding schemes might differ considerably from that of most other eukaryotes.”

Reviewer #2 (Remarks to the Author):

The authors have addressed all my comments. The new data and discussions on canonical and non-noncanonical are very interesting. I would be happy to see this work published.

Response: We thank this reviewer for the positive review.

If I may I want to bring up a few minor points:

1. Figure 1a, middle panel. Almost all species except for human, mouse and fruit fly contains a large fraction of mRNAs without 5' UTR. It is quite unusual, since majority of translation rely on a scanning process in 5' UTR to start translation. Is this due to a lack of accurate annotations on 5' UTR?

Response: Thanks for raising this concern. We agree with this reviewer that many non-model organisms lack accurate annotations of 5' UTRs. We rephrased the related sentence (Lines 115-117, Page 5), which reads as follows:

“The number of uAUGs varied wildly across species, either due to the differences in the sequencing coverage of genomes, the accuracy and completeness of 5' UTR annotation, the number of protein-coding genes, the length of 5' UTRs, or mutational bias in 5' UTRs.”

We also emphasized this point in the legend of Fig. 1a with the following sentence (Lines 1,078-1,079, Page 28): “The unavailability of annotated 5' UTRs for many genes in less-studied organisms is presumably caused by the lack of accurate annotations.”

2. Supplementary Figure 2. It is interesting that the ATG-like triplets (e.g. ATT, TTG, ATC, GTG) are over-represented in 5' UTR, compared with 3' UTR. Does that indicate that non-canonical uORFs are more preferred within 5' UTR? Or alternatively, is this a consequence of depletion of canonical uORFs, since a single mutation on ATG triplets can easily lead to a switch of canonical uORFs to non-canonical uORFs (ATG -> TTG). As shown in this study, non-canonical uORFs are less effective to repress CDS translation, therefore, these single mutations are not removed during evolution.

Response: These comments are enlightening. In this revised version, we discussed these two possibilities with the following sentences (Lines 129-134, Page 5):

“Interestingly, some AUG-like triplets (e.g., AUU, UUG, AUC, and GUG) tended to have higher O/E ratios in 5' UTRs than in 3' UTRs in all the clades. Such AUG-like triplets were either selectively maintained in 5' UTRs as they can be used as noncanonical start codons, or alternatively, were the consequence of the depletion of uAUGs because point mutations can easily convert AUG to AUG-like triplets (e.g., from AUG → UUG) in the 5' UTRs. However, further studies are required to separate these two possibilities.”

3. Rebuttal letter, line 466. “we found that approximately 70-90% of canonical uORFs can be translated (Supplementary Table 3).” It is an interesting data, but I can not find this dataset in Supp. Table 3, or am I missing something? By the way, how to define a translatable uORF, by using Ribo-seq or TIS-seq. Was that done based on a threshold or by using other ORF prediction tools (e.g. PMID: 26657557)?

Response: We apologize for the typo. The summary statistics of translated uORFs were presented in Supplementary Table 4, not Supplementary Table 3. We didn't use any ORF prediction tools to define the translatable uORFs. We defined a translated uORF based on a threshold of ribosome-protected fragments (RPFs) whose P-sites are located in this uORF.

In this revised version, we clarified this point in the Methods section with the following sentence (Lines 570-571, Page 17): "A uORF was considered as translated if it was covered by the P-site of at least one RPF read across different ribosome profiling datasets in a species."

We also provided the list of translated uORFs and associated RPF counts in the Source Data of this revised manuscript (figshare doi: [10.6084/m9.figshare.12612068](https://doi.org/10.6084/m9.figshare.12612068)).

References

- Aspden, J.L., Eyre-Walker, Y.C., Phillips, R.J., Amin, U., Mumtaz, M.A.S., Brocard, M., and Couso, J.-P. (2014). Extensive translation of small Open Reading Frames revealed by Poly-Ribo-Seq. *eLife* 3, e03528.
- Bachvaroff, T.R. (2019). A precedented nuclear genetic code with all three termination codons reassigned as sense codons in the syndinean *Amoebophrya* sp. ex *Karlodinium veneficum*. *PLoS One* 14, e0212912.
- Baranov, P.V., Atkins, J.F., and Yordanova, M.M. (2015). Augmented genetic decoding: global, local and temporal alterations of decoding processes and codon meaning. *Nat Rev Genet* 16, 517-529.
- Brar, G.A., Yassour, M., Friedman, N., Regev, A., Ingolia, N.T., and Weissman, J.S. (2012). High-resolution view of the yeast meiotic program revealed by ribosome profiling. *Science* 335, 552-557.
- Burki, F., Roger, A.J., Brown, M.W., and Simpson, A.G.B. (2020). The New Tree of Eukaryotes. *Trends Ecol Evol* 35, 43-55.
- Calviello, L., Mukherjee, N., Wyler, E., Zauber, H., Hirsekorn, A., Selbach, M., Landthaler, M., Obermayer, B., and Ohler, U. (2016). Detecting actively translated open reading frames in ribosome profiling data. *Nat Methods* 13, 165-170.
- Calvo, S.E., Pagliarini, D.J., and Mootha, V.K. (2009). Upstream open reading frames cause widespread reduction of protein expression and are polymorphic among humans. *Proc Natl Acad Sci U S A* 106, 7507-7512.
- Chen, J., Brunner, A.D., Cogan, J.Z., Nuñez, J.K., Fields, A.P., Adamson, B., Itzhak, D.N., Li, J.Y., Mann, M., Leonetti, M.D., *et al.* (2020). Pervasive functional translation of noncanonical human open reading frames. *Science* 367, 1140-1146.
- Chew, G.L., Pauli, A., and Schier, A.F. (2016). Conservation of uORF repressiveness and sequence features in mouse, human and zebrafish. *Nat Commun* 7, 11663.
- Coyne, R.S., Hannick, L., Shanmugam, D., Hostetler, J.B., Brami, D., Joardar, V.S., Johnson, J., Radune, D., Singh, I., Badger, J.H., *et al.* (2011). Comparative genomics of the pathogenic ciliate *Ichthyophthirius multifiliis*, its free-living relatives and a host species provide insights into adoption of a parasitic lifestyle and prospects for disease control. *Genome Biol* 12, R100.
- Heaphy, S.M., Mariotti, M., Gladyshev, V.N., Atkins, J.F., and Baranov, P.V. (2016). Novel Ciliate Genetic Code Variants Including the Reassignment of All Three Stop Codons to Sense Codons in *Condylostoma magnum*. *Molecular biology and evolution* 33, 2885-2889.
- Johnstone, T.G., Bazzini, A.A., and Giraldez, A.J. (2016). Upstream ORFs are prevalent translational repressors in vertebrates. *The EMBO journal* 35, 706-723.
- Lobanov, A.V., Heaphy, S.M., Turanov, A.A., Gerashchenko, M.V., Pucciarelli, S., Devaraj, R.R., Xie, F., Petyuk, V.A., Smith, R.D., Klobutcher, L.A., *et al.* (2017). Position-dependent termination and widespread obligatory frameshifting in *Euplotes* translation. *Nat Struct Mol Biol* 24, 61-68.

- McGillivray, P., Ault, R., Pawashe, M., Kitchen, R., Balasubramanian, S., and Gerstein, M. (2018). A comprehensive catalog of predicted functional upstream open reading frames in humans. *Nucleic acids research* *46*, 3326-3338.
- Niu, R., Zhou, Y., Zhang, Y., Mou, R., Tang, Z., Wang, Z., Zhou, G., Guo, S., Yuan, M., and Xu, G. (2020). uORFlight: a vehicle toward uORF-mediated translational regulation mechanisms in eukaryotes. *Database (Oxford)* *2020*.
- Orr, M.W., Mao, Y., Storz, G., and Qian, S.-B. (2019). Alternative ORFs and small ORFs: shedding light on the dark proteome. *Nucleic Acids Research*.
- Sengupta, S., and Higgs, P.G. (2015). Pathways of Genetic Code Evolution in Ancient and Modern Organisms. *J Mol Evol* *80*, 229-243.
- Swart, E.C., Serra, V., Petroni, G., and Nowacki, M. (2016). Genetic Codes with No Dedicated Stop Codon: Context-Dependent Translation Termination. *Cell* *166*, 691-702.
- Whiffin, N., Karczewski, K.J., Zhang, X., Chothani, S., Smith, M.J., Evans, D.G., Roberts, A.M., Quaipe, N.M., Schafer, S., Rackham, O., *et al.* (2020). Characterising the loss-of-function impact of 5' untranslated region variants in 15,708 individuals. *Nat Commun* *11*, 2523.
- Záhonová, K., Kostygov, A.Y., Ševčíková, T., Yurchenko, V., and Eliáš, M. (2016). An Unprecedented Non-canonical Nuclear Genetic Code with All Three Termination Codons Reassigned as Sense Codons. *Curr Biol* *26*, 2364-2369.

Reviewers' Comments:

Reviewer #1:

Remarks to the Author:

The authors addressed all my comments in full. I would like to congratulate the authors on the comprehensive and timely study dedicated to the important topic of the uORFs evolution.

Pavel Baranov.

Point-to-point response

Reviewer #1 (Remarks to the Author):

The authors addressed all my comments in full. I would like to congratulate the authors on the comprehensive and timely study dedicated to the important topic of the uORFs evolution.

Pavel Baranov.

Response: We thank this reviewer for the positive feedback.